# Glacial runoff buffers droughts through the 21st century

Lizz Ultee[1], Sloan Coats[2], and Jonathan Mackay[3,4]

[1]Department of Geology, Middlebury College, Middlebury, VT, USA
[2]Department of Earth Sciences, University of Hawaii at Mānoa, Honolulu, HI, USA
[3]British Geological Survey, Environmental Science Centre, Keyworth, UK
[4]School of Geography, Earth and Environmental Sciences, University of Birmingham, Edgbaston, UK

**Correspondence:** L. Ultee (eultee@middlebury.edu)

**Abstract.** Global climate model projections suggest that 21st century climate change will bring significant drying in the terrestrial midlatitudes. Recent glacier modeling suggests that runoff from glaciers will continue to provide substantial freshwater in many drainage basins, though the supply will generally diminish throughout the century. In the absence of dynamic glacier ice within global climate models (GCMs), a comprehensive picture of future hydrological drought conditions in glaciated regions has been elusive. Here, we leverage the results of existing GCM simulations and a global glacier model to evaluate glacial buffering of droughts in the Standardized Precipitation-Evapotranspiration Index (SPEI). We compute one baseline version of SPEI and one version modified to account for glacial runoff changing over time, and we compare the two for each of 56 large-scale glaciated basins worldwide. We find that accounting for glacial runoff tends to increase multi-model ensemble mean SPEI and reduce drought frequency and severity, even in basins with relatively little ($< 2\%$) glacier cover. When glacial runoff is included in SPEI, the number of droughts is reduced in 40 of 56 basins and the average drought severity is reduced in 53 of 56 basins, with similar counts in each time period we study. Glacial drought buffering persists even as glacial runoff is projected to decline through the 21st century. Results are similar under RCP 4.5 and 8.5 emissions scenarios, though results for the higher-emissions RCP 8.5 scenario show wider multi-model spread (uncertainty) and greater incidence of decreasing buffering later in the century. A k-means clustering analysis shows that glacial drought buffering is strongest in moderately glaciated, arid basins.

## 1  Introduction

Runoff from glaciers is an important contributor to water resources in many regions worldwide. For example, runoff from mountain glaciers can account for a significant proportion of dry-season water supply in arid regions (Vergara et al., 2007; Soruco et al., 2015; Pritchard, 2019; Ayala et al., 2020). Further, this glacial runoff can reduce interannual variance in water availability, an effect known as glacial drought buffering (Fountain and Tangborn, 1985; Fleming and Clarke, 2005). There have been substantial recent efforts to project 21st-century changes in glacial runoff at global (Bliss et al., 2014; Huss and Hock, 2018; Marzeion et al., 2018; Cáceres et al., 2020) and regional scales (Juen et al., 2007; Immerzeel et al., 2012; Schaefli et al., 2019; Brunner et al., 2019; Mackay et al., 2019). To understand how these changes will translate to changing basin-scale

water availability and drought buffering, however, requires the added context of regional hydroclimate variability and change
(Kaser et al., 2010).

The use of state-of-the-art global climate models (GCMs) to project hydroclimate change is appealing because the simulated changes reflect self-consistent climate physics on the global-to-regional scale. GCM projections suggest that on large scales the terrestrial midlatitudes will experience significant drying over the coming century (Cook et al., 2014, 2020), adding urgency to the effort to understand future glacial runoff in the context of drought. Moreover, the importance of glacial runoff for water supply varies with regional climate (Kaser et al., 2010; Immerzeel et al., 2010; Rowan et al., 2018), emphasising the need for a holistic view of glaciated-basin hydroclimate change. Unfortunately, there remain uncertainties related to the choice of hydroclimate metric and the role of land surface process in driving projected hydroclimate changes (Milly and Dunne, 2016; Swann et al., 2016; Scheff et al., 2017; Mankin et al., 2018; Yang et al., 2019; Mankin et al., 2019; Ault, 2020), as well as uncertainties stemming from the lack of interactive glacier ice in models. In particular, current global climate models do not account for changing glacier volume and extent, with important consequences for projections of future water availability in glaciated regions (Barnett et al., 2005). Future glacial runoff depends on nonlinear glacier-dynamic response to changing climate (Huss and Hock, 2018; Marzeion et al., 2020), which cannot be simulated directly in GCMs nor extrapolated from observations.

In many cases, GCM land components are not equipped to handle the hydrology of glaciated drainage basins on the century scale. The MATSIRO land surface model (Takata et al., 2003) used in MIROC-ESM, for example, handles water routing through snowpack, but not multiannual storage in glacier ice. The land surface scheme of CNRM-CM6 allows limited water storage in snow and ice and includes a "permanent snow/ice" land tile classification (Decharme et al., 2019), but cannot resolve changes in ice cover over time. The Community Land Model (CLM), used in GCMs including CCSM and NorESM to simulate land-surface dynamics and hydrology, includes glacier ice among its land-cover types but does not account for glacier dynamics or change over time (Lawrence et al., 2018). Further, the spatial resolution of current GCMs leaves them poorly equipped to handle precipitation gradients in high-relief areas (Flato et al., 2013), where mid-latitude glaciers are most likely to be found. Global glacier models have demonstrated that glacier coverage worldwide cannot be assumed static over the coming century (Huss and Hock, 2018; Marzeion et al., 2018, 2020); thus, surface hydrology schemes that do not account for changing glacial water storage over time risk under- or over-estimating the true water availability (van de Wal and Wild, 2001).

A further difficulty is that the climate physics simulated by each GCM are an uncertain approximation of those in the real world. Intercomparisons of model output from multiple GCMs allow for a quantification of the range of projections that result from the uncertain approximations made by each—so called structural uncertainty. These quantifications are hindered, however, by the incomparability of directly-simulated hydroclimate quantities (such as soil moisture) across GCMs. For example, the land components of GCMs range widely in complexity, including different numbers of soil levels with inconsistent corresponding depths (e.g. Cook et al., 2014) and widely varying runoff sensitivities (e.g. Lehner et al., 2019). The resulting difficulty in comparing hydroclimate metrics directly across GCMs has led to the widespread use of offline hydroclimate metrics when quantifying hydroclimate change, specifically in the form of standardized drought indices that facilitate like-for-like intercomparison.

Among the drought indices in operational use (reviewed in World Meteorological Organization and Global Water Partnership, 2016), only a few are globally intercomparable, scalable for different types of drought, and applicable under a variety of future climate change scenarios. For example, the widely-used Palmer Drought Severity Index (PDSI; Palmer, 1965) has a single inherent timescale of approximately nine months, which limits its applicability to certain types of drought. The Standardized Precipitation Index (SPI; McKee et al., 1993) is more flexible, but its lack of consideration for atmospheric moisture demand limits its applicability to future climate change. The Standardized Precipitation-Evapotranspiration Index (SPEI; Vicente-Serrano et al., 2009) satisfies all of the above criteria and offers a user-defined temporal scale to facilitate studies of hydroclimate variability across timescales and climate system components (e.g. Lorenzo-Lacruz et al., 2010; Potop et al., 2012; Kingston et al., 2014; Ault, 2020). SPEI is regularly computed at the coarse spatial resolutions typical of GCMs, both for operational drought monitoring and forecasting and for projections of drought conditions in a changing climate (Cook et al., 2014).

Here, we modify SPEI to quantify glacial drought buffering for all 56 large-scale glaciated drainage basins (hereinafter "basins") worldwide. We bring together glacial runoff simulated in a global glacier model (Huss and Hock, 2018) forced by eight GCMs, with additional hydroclimate variables from the same GCMs, to contextualize the evolving importance of glacial runoff for drought in each basin. We explain the nuanced signature of glacial drought buffering in our modified SPEI as contrasted with a baseline version computed without glacial runoff. Finally, we quantify the characteristics of basins where glacial drought buffering, as expressed in our modified SPEI, is largest or expected to change the most over the coming century.

## 2 Methods

We aim to assess the glacial drought buffering effect on an operational drought index, with applications in projecting future drought conditions. We calculate two versions of SPEI, one accounting for glacial runoff and one ignoring it, and compare them. We choose to analyse SPEI because of its flexibility, as described above, and ready data availability. The Standardized Runoff Indicator (SRI, see Shukla and Wood, 2008), which would seem a natural choice for this study of glacial runoff effects, is not suitable due to uneven data quality and availability. However, we note that SRI has a moderate to strong positive correlation with SPEI where data is available for both indices (see Appendix C), so our results would be similar whether computing SRI or SPEI.

SPEI is a simple climatic water balance, with water accumulation through precipitation and loss through potential evapotranspiration (PET, calculated here following Allen et al., 1998), normalized such that its mean over a reference period is 0 and its standard deviation is 1. SPEI $< 0$ corresponds to drier conditions and SPEI $> 0$ to wetter conditions. For more detailed SPEI methodology, please see Appendix B. Our approach isolates the glacial effect on SPEI using hydroclimate output of eight GCMs combined with offline simulated glacial runoff (Huss and Hock, 2018) forced by boundary conditions from the same GCMs. A particular strength of the SPEI approach is that it can be calculated over different integration timescales (typically between 1-48 months) which allows for the quantification of particular water stores of interest (e.g. soil moisture/streamflow/groundwater). This integration of meteorological forcing recognises that the terrestrial hydrological system

acts as a low pass filter that can result in significant pooling, lengthening, attenuation and lag of meteorological drought, especially for long residence-time water stores (van Loon, 2015). Given that this study aims to evaluate glacial runoff drought-buffering, we use a relatively short 3-month integration timescale, which is typical of that used to assess streamflow drought

(López-Moreno et al., 2013; Peña-Gallardo et al., 2019). In choosing this timescale we assume that the majority of glacial runoff passes through the basin via relatively fast overland and shallow-subsurface flow pathways. Results for timescales between 3 and 27 months are available in our public repository for the reader interested in other types of drought or timescales of hydroclimate variability.

We leverage existing glacial runoff estimates generated by Huss and Hock (2018) for all large-scale ($> 5000$ km$^2$) drainage

basins in which present glacier ice coverage is at least 30 km$^2$ total and at least $0.01\%$ of basin area. There are 56 such basins outside of Greenland and Antarctica. They comprise 16 basins in Asia, 11 in Europe, 16 in North America, 12 in South America, and 1 in New Zealand. Table A1 lists all basins studied and their key characteristics. Maps of basin location and projected change in glacial runoff appear in Huss and Hock (2018).

We identify eight GCMs that *(i)* provide the variables necessary to calculate SPEI and *(ii)* have a corresponding glacial runoff

projection from Huss and Hock (2018). Those GCMs are the following CMIP5 participants: CanESM2, CCSM4, CNRM-CM5, CSIRO-Mk3-6-0, GISS-E2-R, INMCM4, MIROC-ESM, and NorESM1-M (Taylor et al., 2011). For each GCM, we select the same representative concentration pathway (RCP) 4.5 and 8.5 simulations (Taylor et al., 2011) that were used to force projections in Huss and Hock (2018). From those GCM simulations, we extract atmospheric surface temperature, surface pressure, total precipitation, surface specific humidity, and surface net radiation for each of the 56 basins we study. Specifically,

we identify all latitude-longitude grid cells from the native GCM grid that fall within the boundary of the basin as defined by the Global Runoff Data Centre (2007), extract the required variables from the associated grid points, and take an area-weighted sum to produce a single, basin-total timeseries for each variable in each basin. We then calculate PET with the basin-total timeseries for each variable using the reference crop approximation of Allen et al. (1998) with the addition of a stomatal conductance term following Yang et al. (2019). We calculate SPEI with the resulting basin-total PET timeseries and the basin-total precipitation

timeseries (see below and Appendix B).

## 2.1 SPEI modified to account for glacial runoff

To test the role of glacial runoff in basin-level water availability as indicated by SPEI, we calculate two versions of the index. The first, SPEI$_N$, is calculated for each GCM following Vicente-Serrano et al. (2009), with modifications for variable stomatal conductance (Yang et al., 2019) and non-parametric standardization (Farahmand and AghaKouchak, 2015) as described in

Appendix B, with no accounting for glacial runoff. For the second, SPEI$_G$, we account for glacier runoff by modifying the moisture source term in the calculation. We replace the total precipitation input $P$ with

$$\tilde{P} = \frac{A - A_g}{A} P + R, \tag{1}$$

where $\tilde{P}$ is the modified moisture source term, $P$ is the initial moisture source term from each GCM, $A_g$ is the initially glaciated area of the basin, $A$ is the total basin area, and $R$ is the glacial runoff for that basin from Huss and Hock (2018)

forced with the same GCM (see Appendix B2). We note that Huss and Hock (2018) apply an elevation-dependent correction to the precipitation input data for their model; to avoid introducing any further assumptions, we have not attempted to apply any similar correction to the precipitation term in our modified SPEI calculation.

## 2.2 Baseline change in water availability

For each basin, we compute and compare the 30-year, multi-GCM ensemble running mean and variance of the $SPEI_N$ and
$SPEI_G$ time series, for the period 1980-2100. We also compare GCM-by-GCM changes in SPEI mean and variance at the end of the 21st century (2070-2100) for RCP 4.5 and 8.5.

## 2.3 Buffering described in statistics of drought in SPEI

Next, we describe glacial drought buffering in terms of its effect on statistics of drought incidence. For each basin, in each single-GCM time series computed with and without glacial runoff, we identify "droughts" as SPEI excursions that exceed a
threshold of $-1$ (as in drought monitoring applications, e.g. Ming et al., 2015; Danandeh Mehr and Vaheddoost, 2020). For each such event, we report the drought severity as the total accumulated SPEI deficit during the continuous negative-SPEI period that includes the excursion (see Figure B3). The units of severity are therefore the same as SPEI itself: a unitless standardized index.

We define glacial drought buffering as the difference in each statistic between the $SPEI_N$ and $SPEI_G$ series. For example,
positive buffering of the number of droughts corresponds to $SPEI_G$ series with fewer droughts in a given period than the $SPEI_N$ series in the same period. Positive buffering of drought severity corresponds to $SPEI_G$ series in which the mean drought severity during the period is less than that in the $SPEI_N$ series.

## 2.4 Change in drought buffering over time

We compute the statistics as described in Section 2.3 for a historical period (1980-2010), the mid-21st century (2030-2060), and
the end of the 21st century (2070-2100). We assess the change in the multi-GCM mean of each statistic for each basin across the three time periods and report whether buffering in future time periods is likely greater or less than that in the historical period. Finally, we compute the Spearman rank-correlation coefficient between the multi-GCM mean of each statistic for each basin and several basin-scale variables that could account for differences in buffering: the percent glacier coverage by area, the total basin area, the historical mean precipitation over the basin, and the aridity index of the basin over the historical period (see
Table 1). The aridity index employed here is the ratio of multi-GCM mean precipitation divided by multi-GCM mean PET, for each basin, over the period 1980-2010.

## 2.5 K-means cluster analysis of basin characteristics that affect buffering

We anticipate that there may be several variables that when considered together control the magnitude of glacial drought buffering in a given basin. We thus explore multi-variate relationships among the four potential explanatory variables. Specifically,

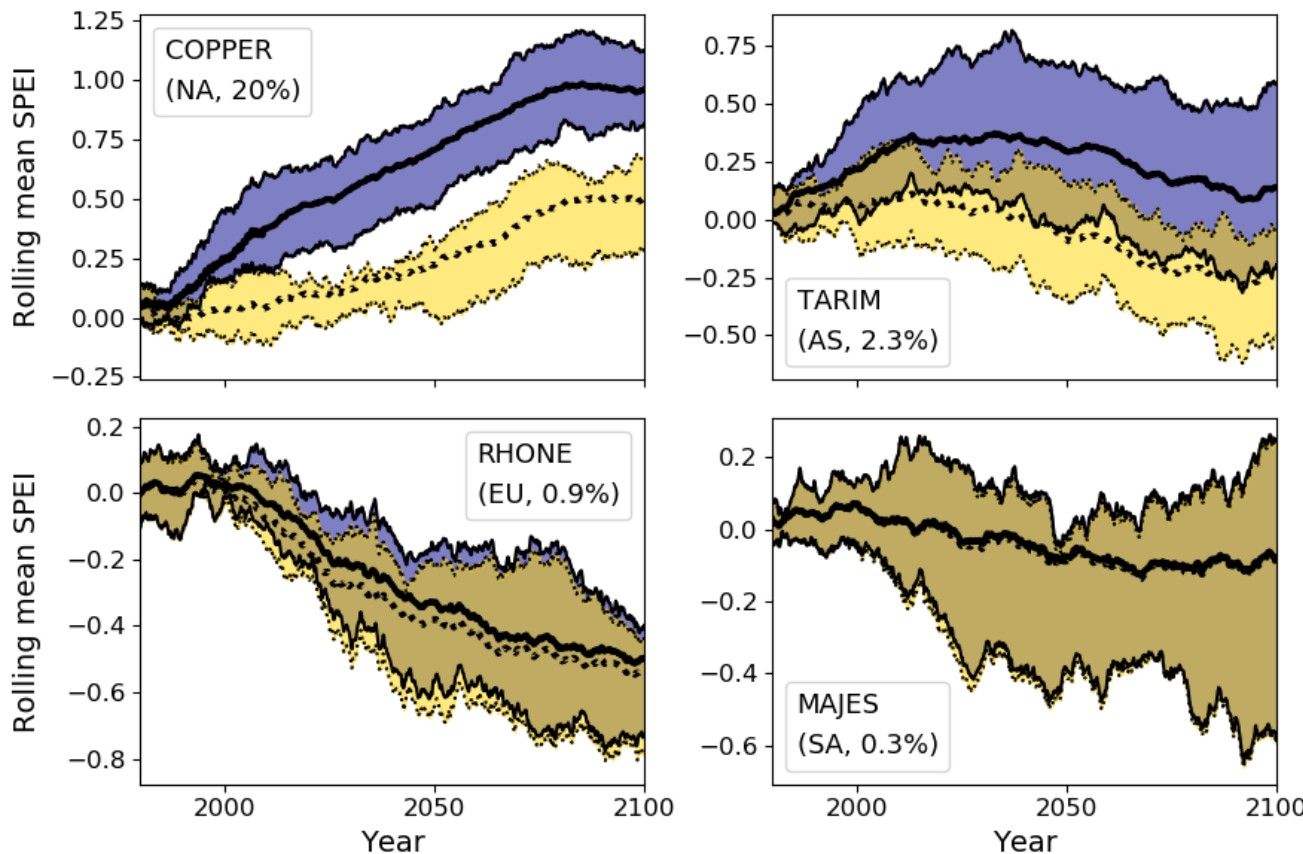

**Figure 1.** 30-year running ensemble mean and interquartile range of SPEI$_{3mo}$ with glacial runoff (blue shades) and without (yellow shades) for the RCP 4.5 scenario in four example basins (name, region, and percent glacier coverage by area in corner of each figure panel). 'NA' indicates North America, 'AS' Asia, 'EU' Europe, and 'SA' South America.

we apply a k-means cluster analysis using a built-in function of the scikit-learn Python package (Pedregosa et al., 2011). This method requires the target number of clusters, $k$, to be given as an input. We tested clustering with $k$ ranging from 2 to 5. Here, we present the results for $k = 2$, which we found preserved most of the character of the results found for higher $k$-values without introducing spurious information. We then explored the differences between clusters using the four potentially explanatory variables described in Section 2.4 above.

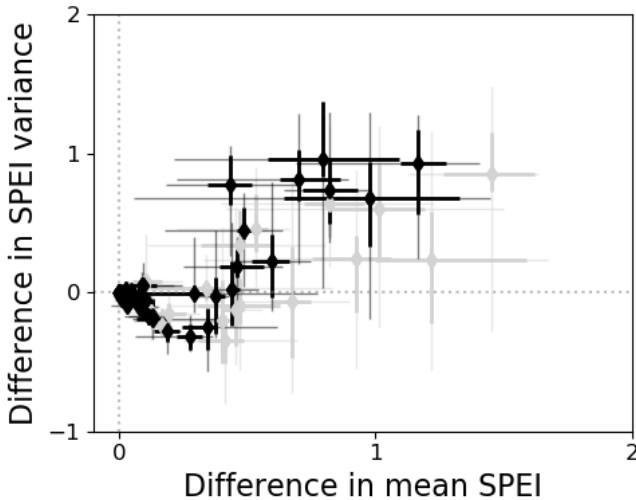

**Figure 2.** Difference due to explicit accounting of glacial runoff in SPEI$_{3mo}$ 30-year mean and variance at end of 21st century (2070-2100), for climate scenarios RCP 4.5 (black) and RCP 8.5 (grey). A diamond marker for each of the 56 basins analysed shows the difference in the ensemble mean SPEI 30-year mean (x-axis) and variance (y-axis) for each basin. Whiskers show the range of single-GCM results for each basin, with interquartile range in heavier lines.

## 3  Results

### 3.1  Glacier runoff changes background water supply

Accounting for glacial runoff as described above increases 30-year rolling ensemble mean SPEI in every basin, regardless of whether the basin as a whole is projected to become wetter or drier throughout the century (see extended results, Appendix A). However, there is considerable variation in the temporal trends of the glacial effect on mean SPEI both across basins and between GCMs in a single basin.

Figure 1 shows the 30-year rolling ensemble mean SPEI for four representative basins. We have chosen to highlight these basins as they are geographically distributed, span the range of basin glacial coverage ($A_g/A$ in Equation 1 above), and have projected future SPEI with both drying and wetting trends; results for all 56 basins appear in Appendix A. In the Copper River basin of Alaska, the ensemble mean and interquartile range show SPEI increasing throughout the 21st century, with even more pronounced increases when glacial runoff is taken into account. In the Rhone basin of central Europe, the ensemble mean projects decreasing SPEI throughout the century to be slightly mitigated by glacial runoff. The ensemble mean results are agnostic about the temporal trend in SPEI for the Majes basin of Peru; none are much changed by the inclusion of glacial runoff, apart from a very slight increase in the 25th percentile of SPEI during 2010-2050. Most interesting is the Tarim basin of central Asia. When glacial runoff is not considered, the ensemble mean projection shows SPEI decreasing throughout the 21st century, becoming negative on average after 2050. However, with glacial runoff included, GCMs show an initial increase in

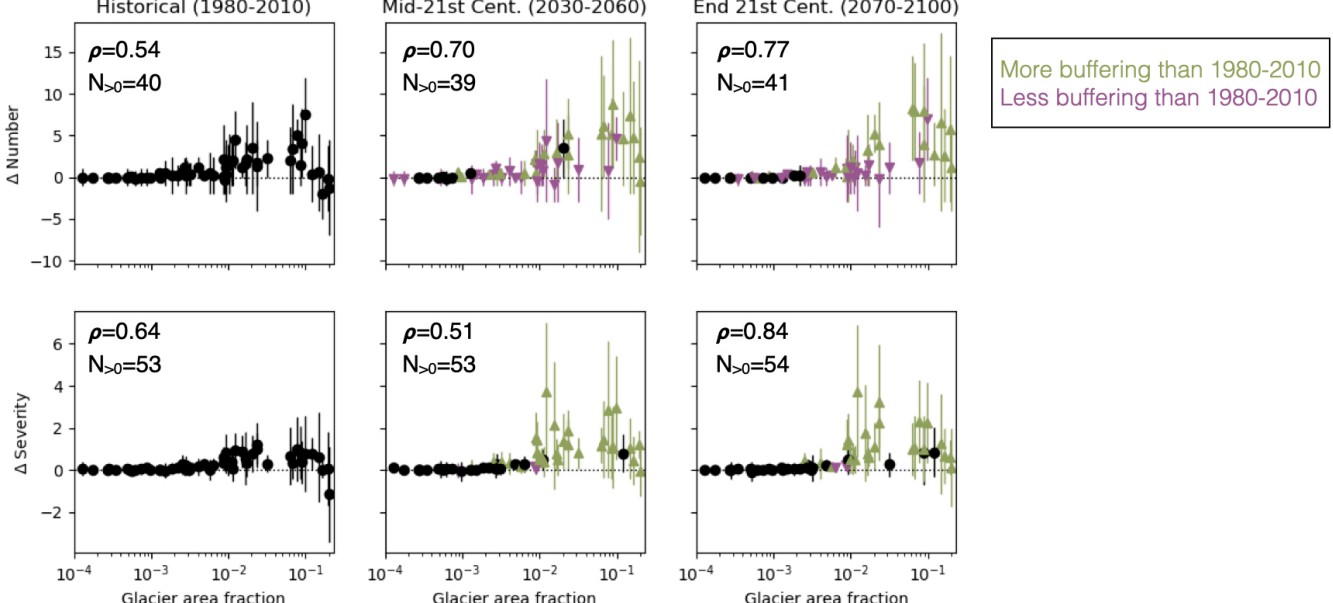

**Figure 3.** Glacial buffering of number of droughts (top row) and drought severity (bottom row), for the periods 1980-2010 (left), 2030-2060 (center), and 2070-2100 (right). Differences are given as the statistic on $SPEI_N$ - $SPEI_G$, so a positive $y$-axis means positive buffering, i.e. there are more, and/or more severe, droughts in the without-glacial runoff series. X-axes are glacier area fraction $A_G/A$ presented on a logarithmic scale. Markers indicate the ensemble mean for each basin and whiskers indicate range among single-GCM results. In the second and third columns, color indicates change in ensemble mean buffering versus the historical period: green upward triangles for increased buffering, black dots for no change (tolerance of $\pm 0.1$), and magenta downward triangles for decreased buffering. Annotations in the corner of each panel list Spearman's rank correlation coefficient $\rho$ between basin glaciated area and the statistic shown in that panel, as well as the number of basins for which multi-model mean buffering is positive, $N_{>0}$. Results shown here pertain to the RCP 4.5 scenario; see results for RCP 8.5 in Figure A2.

SPEI that remains positive (though decreasing) through the end of the century. This suggests that in the Tarim basin, accounting for glacial runoff projections would change the projected future hydroclimate from one with less water availability for human and ecosystem services to one with greater water availability in the 21st relative to the 20th century.

In addition to shifts in mean SPEI, we also identify shifts in SPEI variance. Figure 2 plots the glacial-runoff shift in SPEI mean and variance for the period 2070-2100. There are several basins with moderate increases in mean SPEI and decreases in variance, and a handful of basins with large increases in mean SPEI paired with correspondingly large increases in variance. These changes in variance motivated a closer look at the incidence of droughts.

### 3.2 Including glacial runoff buffers drought frequency and severity

Figure 3 shows the glacial buffering (difference of statistics from $SPEI_N$ versus $SPEI_G$) of number of droughts and drought severity during each of three periods. Over the historical period 1980-2010 (left column of Figure 3), the buffering effect on

| Statistic | % Glaciated | Basin Area | Historical Precip. | Historical AI |
|---|---|---|---|---|
| $\Delta$ Number, hist. | **0.54** | -0.21 | -0.25 | 0.02 |
| $\Delta$ Number, mid-C | **0.70** | -0.30 | -0.34 | 0.34 |
| $\Delta$ Number, end-C | **0.77** | **-0.52** | -0.50 | **0.54** |
| $\Delta$ Severity, hist. | **0.64** | -0.19 | -0.25 | -0.10 |
| $\Delta$ Severity, mid-C | **0.75** | -0.27 | -0.34 | 0.01 |
| $\Delta$ Severity, end-C | **0.84** | -0.29 | **-0.38** | 0.07 |

**Table 1.** Spearman's rank correlation $\rho$ between RCP 4.5 drought statistics shown in Figure 3 (rows) and basin-level variables (columns). Boldface indicates $p < 0.01$. The variables tested here are described in Section 2.4; the abbreviation "AI" indicates the aridity index.

both metrics is generally positive ($\text{SPEI}_G$ has fewer and less severe droughts than $\text{SPEI}_N$) and largest for moderately glaciated basins. Multi-model mean buffering is positive during the historical period in 40 of the 56 basins with regard to number of droughts, and 53 of 56 basins with regard to drought severity ($N_{>0}$ annotations on Figure 3). By the mid-21st century (2030-2060, center column of Figure 3), some basins see decreased buffering of number of droughts, but others see increased buffering. Among basins where there is nontrivial change over time, almost all show stronger buffering of drought severity during the mid-century compared with the historical period. In the end-of-century period (2070-2100, right column of Figure 3), buffering of number and severity of droughts remains generally positive, and some heavily glaciated basins have stronger buffering than they did during the historical period. The number of basins with multi-model mean positive buffering on each metric remains consistent over time (little change in the $N_{>0}$ counts reading from left to right on Figure 3).

For a few basin/model combinations, buffering of drought number or severity appears negative during one or more of the time periods studied. That is, there are more droughts identified in the 3-month $\text{SPEI}_G$ time series than in $\text{SPEI}_N$, or the average drought identified in $\text{SPEI}_G$ is more severe. Negative buffering of both number and severity is rare, appearing in only 7 of 448 basin/model combinations, and the effect is small when present (see Appendix B).

Spearman's rank-correlation coefficient $\rho$ indicates a strong relationship ($\rho > 0.5$, $p < 0.01$) between each basin's glacier area fraction and the ensemble mean of glacial buffering effect on drought number and severity. The rank-correlation relationship is stronger at the end of the 21st century than in the historical period. That is, glacial drought buffering correlates strongly with basin glacier area fraction, and even more so at the end of the century. Other potential explanatory variables, such as total basin area, mean precipitation during the historical period, or aridity index during the historical period, showed weaker and generally less significant correlations with buffering (Table 1).

## 3.3 Drought buffering through the 21st Century is stronger for more heavily glaciated basins

Although basin glacier coverage is an important explanatory factor in the strength of glacial drought buffering over time, each variable shown in Table 1 does have a significant ($p < 0.01$) relationship with a glacial drought buffering metric in at least one time period studied. We explored multi-variate relationships among the four potential explanatory variables with k-means clustering, as described in Section 2.5.

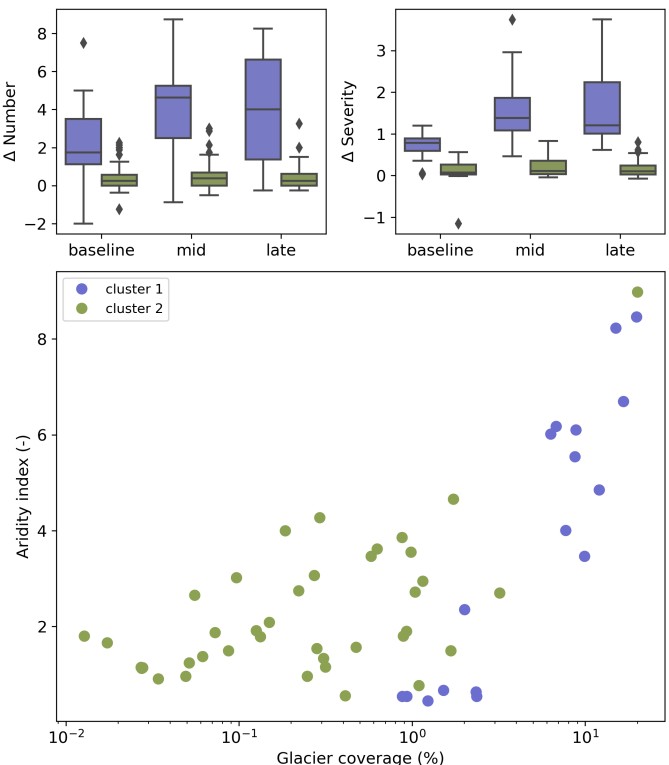

**Figure 4.** k-means clustering of RCP 4.5 glacial-buffering metrics (top row) and scatter plot of basin aridity index and percentage glacier coverage coloured by cluster (bottom). In the box plots of glacial-buffering metrics over time, colored boxes indicate the interquartile range among single-basin results in each cluster, with lines indicating the median, whiskers indicating the range of the full distribution, excluding the outliers identified by the diamonds. In the scatter plot, each dot indicates the ensemble mean of each quantity for a single basin. Recall that aridity index is a ratio $P/PET$ (Section 2.4), such that higher values indicate wetter conditions.

We found two clusters of basins with distinct historical and future glacial-buffering metrics (Figure 4, top row). Basin glacier coverage and aridity index were the principal explanatory variables (Figure 4, bottom). Cluster 1 includes 17 basins which are characterised by higher median levels of glacial buffering and an overall increase in buffering through the 21st century for both drought metrics. This cluster includes heavily glaciated basins (glacier coverage up to 19.7%, high x-axis values in Figure 4) where there is a large quantity of water stored as glacial ice, but also more arid basins (low y-axis values in Figure 4) such as

the Tarim, where glacial runoff is a substantial water source despite modest glacier coverage (2%). Cluster 2 consists of the remaining 39 basins and is characterised by low to moderate levels of glacial buffering for all time periods and drought metrics. All but one of these basins has <3% glacier coverage and, therefore, buffering by glacial runoff is relatively low. Cluster 2 also includes the basin with the largest glacier coverage (Copper, 20%) and highest aridity index (8.98), indicative of large moisture supply relative to demand, and thus suggesting that the drought-buffering effect of a high volume of glacial runoff is diminished

when non-glacial moisture sources in the basin are also large.

## 4 Discussion

Huss and Hock (2018) found that the response of glacial runoff to 20th-21st century climate change took the shape of a bell curve, with maximum basin-level runoff ("peak water") occurring in some year after the onset of glacial retreat. Our analysis of $SPEI_N$ and $SPEI_G$ shows that in most basins, the effect of including glacial runoff is an increase in ensemble mean SPEI that diminishes later in the 21st century (Figure 1 and A1). This pattern is consistent with the "peak water" framing. However, for several basins including the Copper and Tarim (Figure 1), even the end-of-century decline in glacial runoff does not return mean SPEI to values without glacial runoff. That is, the relevance of glaciers for future drought is not limited to this century.

Theoretical understanding suggests that glacial drought buffering moderates interannual variance in water availability for basins with substantial glacial cover (Fountain and Tangborn, 1985; Chen and Ohmura, 1990; Fleming and Clarke, 2005). We find that accounting for glacial runoff decreases variance in SPEI for some basins and increases it for others (Figure 2). The sign and magnitude of the difference in variance appears to correspond with the magnitude of the difference in mean SPEI; that is, larger contributions to baseline water supply could also increase variability compared with the non-glacial case.

The magnitude of change in end-of-century SPEI mean and variance (Figure 2), and the buffering of individual drought statistics shown in Figures 3-4, is related to the basin glacier area fraction but the relationship is complicated by other variables (Table 1 and Section 3.3). Similarly, van Tiel et al. (2020) found that a relationship exists between interannual streamflow variability and catchment glacier area fraction, but that the relationship could vary with other catchment characteristics and that it was not stable over time. These results caution against a simplistic application of glacier simulations alone to interpret future drought buffering capacity.

In most basins and for most GCMs, glacial runoff remains effective in increasing mean SPEI at the end of the century under both RCP 4.5 and 8.5 (Figure 2). Our analysis of the change in drought frequency and severity (Figures 3-4, RCP 4.5; Figure A2, RCP 8.5) also revealed few basins with substantial decline in buffering of both metrics over the course of the 21st century. However, under RCP 8.5, as compared to RCP 4.5, there are more GCMs and basins in which the end-of-century glacial effect on SPEI variance is weak buffering or amplification (negligible or positive y-axis values in Figure 2), as well as more GCM-basin pairs for which buffering of number of droughts decreases over time (magenta markers in Figure A2). Under the RCP 8.5 scenario, the number of basins with positive multi-model mean buffering ($N_{>0}$ annotations, Figure A2) increases in the mid-21st century and then decreases at the end of the century, a pattern not present in the RCP 4.5 results (Figure 3). The inter-GCM range of projected buffering effect on SPEI is also greater under RCP 8.5. We interpret that the greater warming under RCP 8.5 shrinks glaciers more quickly and thereby further reduces seasonally-available meltwater, such that the basin transitions to a precipitation-dependent regime. Such a scenario supports the prediction that glacial drought buffering will decline with unabated 21st century climate change (Biemans et al., 2019). The reliability of glacial drought buffering for the 21st century thus depends on the global choice of emissions scenario.

Our cluster analysis indicates that glacial drought buffering is strongest, and projected to increase over time, in moderately glaciated, arid basins. Cluster 1 (Figure 4), for which median glacial drought buffering effect is higher and increasing over time, includes basins with glacier coverage ranging from 2% to nearly 20% as well as the basins with the lowest aridity indices

(most arid). In arid basins such as the Tarim, the supply of glacial meltwater may not be large, but other water sources are sufficiently small that even limited glacial runoff has a pronounced effect on SPEI within the basin. Previous authors have also commented on the importance of glacial runoff in arid basins (Pritchard, 2019; Ayala et al., 2020) and dry seasons (Soruco et al., 2015; Frans et al., 2016; Biemans et al., 2019).

In the context of current glacier-modelling efforts that show glacial runoff decreasing with continued climate change (Juen
et al., 2007; Immerzeel et al., 2010; Marzeion et al., 2018; Huss and Hock, 2018), it has not previously been apparent that glaciers will continue to buffer droughts through the end of the 21st century. In qualitative assessments, both Rowan et al. (2018) and Pritchard (2019) found that current glacier meltwater production is unsustainably high in high-mountain Asia and that the glacial fraction of downstream runoff is likely to decline over the 21st century. Immerzeel et al. (2020) found that water stored in glaciers is an important resource of mountain "water towers" worldwide, and assessed that several glaciated
basins are vulnerable to future change. Most recently, an observational study by Hugonnet et al. (2021) found that mass loss from glaciers worldwide has accelerated since the start of the 21st century. However, each of these studies makes only indirect connections between future changes in glacial runoff and the additional hydroclimate processes that will shape future drought. Our SPEI-based analysis incorporates these previous findings in glacial runoff modelling and adds the basin-level hydroclimate context necessary to interpret future glacial drought buffering in a changed climate.

The simple offline computation we present here helps account for the first-order glaciological effect on future basin-level water availability for human and ecosystem services. However, offline computations are unable to capture atmospheric feed-backs of changing mountain glacier extent. For example, ice and snow-covered surfaces reflect more incident radiation to the atmosphere than bare rock or soil surfaces do. One recent study indicated that drought in the extratropical Andes served to re-duce glacier albedo, further enhancing meltwater production (Shaw et al., 2021). Further, water vapor sublimated from glacier
ice or evaporated from supraglacial meltwater pools is a ready source of moisture to the local atmosphere. Finally, glacier surfaces are favorable for creation of strong downslope (katabatic) winds, which can be the dominant feature in local-scale atmospheric circulation (e.g. Obleitner, 1994; van den Broeke, 1997; Aizen et al., 2002). To the extent that any of these local processes are parameterized in current GCMs, their projection into the future will suffer from the inaccurate assumption that glacier ice cover is permanent. The effects of these feedbacks will only be resolved with eventual addition of coupled mountain
glacier schemes in GCMs.

Our offline computation method also comes with the caveat that it is likely to capture effects that are not strictly glacial. For example, in Equation 1, we scale the basin-total precipitation by the total non-glaciated area rather than scaling down precipitation from the specific GCM grid cells where glaciers are found. This methodological choice may tend to overestimate the moisture source term when glaciers are found in comparatively wet parts of a basin (see Section B2.1). Further, because
glacial runoff consists of both meltwater from glacier ice as well as precipitation falling on and then running off of glaciers, a consistent comparison of past and future runoff from a currently glacierized basin requires a catchment outline that does not change over time. The runoff output from Huss and Hock (2018) tracks all water discharge from a constant catchment for each glacier simulated, such that snow and rain falling on areas from which glaciers retreat during the 21st century will still be counted as "glacier" runoff (see Appendix B2). A glacier catchment where the initial glacier has vanished would produce

no glacial melt, but would still produce runoff comprised of precipitation. In principle, this detail is unlikely to affect our results, since we correct the SPEI moisture source term to avoid double-counting (Equation 1). However, Huss and Hock (2018) also apply a precipitation enhancement factor $c_{prec}$ in their model to account for underestimation of high-elevation precipitation in GCM-derived datasets. The value of $c_{prec}$ is calibrated per glacier starting from a default of 1.5 (see Huss and Hock, 2015, Equation 2 and Section 4). That is, for a catchment where the initial glacier has vanished, GCM-derived precipitation input would still be scaled up by a factor of $c_{prec}$. This scaling may produce $SPEI_G > SPEI_N$ even when the glaciers in a catchment have completely receded. For the most heavily glaciated basin in our analysis, the Copper basin at 20% glaciated, scaling up precipitation by the default $c_{prec}$ value over all glaciated area would result in a 10% increase in total basin precipitation. Thus, the drought buffering effects we present here may be in part attributed to the efforts by Huss and Hock (2015) to capture orographic precipitation enhancement, rather than coming from glacial meltwater alone. We believe that this is in fact an additional benefit of accounting for runoff from glaciated regions in greater detail than the current generation of GCMs permits. A sensitivity analysis (Section B2.1; Table B1) indicates that non-glacial enhancement of the moisture source term in SPEI can produce strong drought buffering, but the effect is distinct from the drought buffering calculated from glacial runoff. More detailed analyses to partition the two effects will improve future forecasts of glacial drought buffering.

Here, we have focused on global intercomparison of future basin-level water availability and drought buffering. Local-level water resource studies may require more granular information (Milly et al., 2008; Head et al., 2011; Frans et al., 2016). Our method can be adapted for use with regional climate models (e.g. Noël et al., 2015; Skamarock et al., 2019), with models simulating individual glacier evolution (e.g. Gagliardini et al., 2013; Maussion et al., 2019; Rounce et al., 2020), and in probabilistic ensemble simulations. The multiple temporal horizons of SPEI also make our method scalable, allowing analyses of different types of droughts and supporting eventual integrated physical-socioeconomic studies of the impacts of glacier change (Carey et al., 2017).

## 5 Conclusions

Basin-level water availability as observed and experienced in the present is affected by numerous regionally-variable factors, including the supply of water from glaciers. GCMs in use to study past and future hydroclimate are ill-equipped to capture decade-to-century scale variation in glacial runoff. Although fully dynamic representations of glacier ice within GCMs will be necessary to produce a physically consistent projection of hydroclimate change in glaciated basins, we have presented a simple method to leverage existing glacier model developments (Huss and Hock, 2018) and account for changing glacial runoff in 21st-century projections of hydrological drought. Our analysis shows that applying glacier model output to account for glacial runoff in SPEI tends to increase mean SPEI and buffer drought frequency and severity throughout the 21st century, even in basins with $< 2\%$ glaciation by area. As glaciers continue to retreat late in the century, their "drought buffering" effect on SPEI diminishes but does not vanish. Drought buffering persists under both RCP 4.5 and RCP 8.5 emissions scenarios, but with greater uncertainty and more decline over time in the RCP 8.5 scenario. Thus, the reliability of glacial drought buffering in the 21st century depends on whether the world acts to mitigate greenhous gas emissions. What's more, glacial drought buffering on

SPEI shows wide inter-GCM spread, suggesting considerable structural uncertainty. More fundamental work on the modelling of hydroclimate is thus clearly needed. Of greatest relevance to hydroclimate in glaciated basins will be the inclusion of online glacier models, increasing model resolution and associated improvements in the representation of hydroclimate-topography interactions, and improved simulation of frozen precipitation processes.

*Code and data availability.* Code, data, and a Jupyter notebook guide are available at https://doi.org/10.5281/zenodo.5711935.

## Appendix A: Extended results

All analysis shown in the main text is reproducible and extensible for any of the 56 basins using the Jupyter notebook and code we have provided on GitHub (see Code and Data Availability Statement). We encourage readers interested in detailed results for a specific basin to make use of the material provided there. The public code also allows users to examine SPEI under the RCP 8.5 rather than the RCP 4.5 emissions scenario, and to change timescales of analysis. For example, Jupyter notebook users can choose to view running means over 5-year windows rather than the 30-year windows shown in Figure 1 or analyse SPEI with integration timescales up to 27 months (Appendix B1).

For readers' convenience, we include below extended results for each of the 56 basins we analyzed, for the same timescales and climate scenario shown in the main text. Figure A1 shows the glacial effect on 30-year rolling ensemble mean SPEI, for all 56 basins rather than the 4 example basins shown in Figure 1 . The panels in both figures were computed with the RCP 4.5 emissions scenario, examining SPEI with a 3-month integration timescale. The accompanying Table A1 provides basin location, total area, glacier coverage, and aridity index. Finally, Figure A2 shows the change in drought buffering over time as in Figure 3, but for RCP 8.5 rather than 4.5.

Table A1: Characteristics of the 56 basins studied, arranged in descending order of total glaciated area (matching Figure A1). Total area $A$ is based on the Global Runoff Data Centre (2007) basin outlines. Glaciated area $A_g$ per basin was provided by Matthias Huss, based on Randolph Glacier Inventory version 4.0 (RGI Consortium, 2014, updated 2017). Aridity index is a multi-GCM mean over the 1980-2010 historical period.

| Basin Name | Continent | Central lat/lon (° N, ° E) | Total area $A$ [km$^2$] | Glaciated area $A_g$ [km$^2$] | Aridity index |
|---|---|---|---|---|---|
| Indus | Asia | (31, 74) | 1139075 | 26893.8 | 0.54 |
| Tarim | Asia | (37, 81) | 1051731 | 24645.4 | 0.64 |
| Brahmaputra | Asia | (28, 90) | 518011 | 16606.7 | 2.70 |
| Aral Sea | Asia | (42, 67) | 1233148 | 15176.7 | 0.45 |
| Copper | N. America | (61, -143) | 64959 | 12998.0 | 8.98 |
| Ganges | Asia | (25, 82) | 1024462 | 11216.0 | 0.78 |
| Yukon | N. America | (63, -144) | 829632 | 9535.4 | 2.95 |

| Alsek | N. America | (60, -137) | 28422 | 5614.8 | 8.46 |
|---|---|---|---|---|---|
| Susitna | N. America | (62, -149) | 49470 | 4304.0 | 5.54 |
| Balkhash | Asia | (45, 78) | 423657 | 3945.4 | 0.55 |
| Stikine | N. America | (57, -129) | 51147 | 3467.6 | 6.18 |
| Santa Cruz | S. America | (-49, -71) | 30599 | 3027.8 | 3.47 |
| Fraser | N. America | (52, -122) | 239678 | 2495.1 | 2.72 |
| Baker | S. America | (-46, -72) | 30760 | 2372.3 | 4.00 |
| Yangtze | Asia | (30, 106) | 1745094 | 2317.4 | 1.79 |
| Salween | Asia | (25, 96) | 258475 | 2295.9 | 1.80 |
| Columbia | N. America | (46, -116) | 668561 | 1878.4 | 1.54 |
| Issyk-Kul | Asia | (42, 73) | 191032 | 1677.3 | 0.55 |
| Amazon | S. America | (-6, -64) | 5880854 | 1634.1 | 1.14 |
| Colorado | S. America | (-35, -67) | 390631 | 1601.2 | 0.56 |
| Taku | N. America | (58, -132) | 17967 | 1583.6 | 6.10 |
| Mackenzie | N. America | (61, -120) | 1752001 | 1519.2 | 1.50 |
| Nass | N. America | (56, -129) | 21211 | 1337.3 | 6.02 |
| Thjórsá | Europe | (64, -19) | 7527 | 1251.8 | 6.70 |
| Joekulsá | Europe | (65, -16) | 7311 | 1098.6 | 8.23 |
| Kuskokwim | N. America | (61, -156) | 118114 | 1032.8 | 3.86 |
| Rhone | Europe | (45, 5) | 97485 | 904.2 | 1.90 |
| Skeena | N. America | (55, -127) | 42944 | 742.3 | 4.66 |
| Ob | Asia | (55, 75) | 2701040 | 739.5 | 1.13 |
| Oelfusá | Europe | (64, -20) | 5678 | 683.4 | 4.85 |
| Mekong | Asia | (22, 101) | 787256 | 485.7 | 1.37 |
| Danube | Europe | (46, 18) | 793704 | 408.4 | 1.25 |
| Nelson River | N. America | (51, -101) | 1099380 | 374.7 | 0.91 |
| Po | Europe | (45, 9) | 73066 | 347.3 | 1.57 |
| Kamchatka | Asia | (55, 159) | 54103 | 312.7 | 3.47 |
| Rhine | Europe | (49, 7) | 190522 | 285.0 | 2.09 |
| Gloma | Europe | (61, 10) | 42862 | 269.4 | 3.62 |
| Huang He | Asia | (36, 107) | 988062 | 267.9 | 1.15 |
| Indigirka | Asia | (67, 144) | 341227 | 248.4 | 1.88 |
| Lule | Europe | (66, 18) | 25127 | 247.2 | 3.56 |

| Rapel | S. America | (-34, -70) | 15689 | 238.1 | 0.67 |
|---|---|---|---|---|---|
| Santa | S. America | (-8, -77) | 11882 | 198.9 | 1.50 |
| Skagit | N. America | (48, -121) | 7961 | 159.5 | 2.36 |
| Kuban | Asia | (44, 40) | 58935 | 146.0 | 0.96 |
| Titicaca | S. America | (-16, -68) | 107215 | 134.5 | 1.91 |
| Nushagak | N. America | (60, -156) | 29513 | 86.4 | 4.28 |
| Biobio | S. America | (-37, -71) | 24108 | 76.2 | 1.15 |
| Irrawaddy | Asia | (23, 96) | 411516 | 71.2 | 1.66 |
| Negro | S. America | (-39, -68) | 130062 | 64.1 | 0.96 |
| Majes | S. America | (-15, -72) | 18612 | 57.3 | 1.34 |
| Clutha | New Zealand | (-45, 169) | 17118 | 46.5 | 3.07 |
| Daule-Vinces | S. America | (-1, -79) | 41993 | 40.6 | 3.02 |
| Kalixaelven | Europe | (67, 20) | 17157 | 37.9 | 2.75 |
| Magdalena | S. America | (6, -74) | 261204 | 33.3 | 1.80 |
| Dramselv | Europe | (60, 9) | 17364 | 32.1 | 4.00 |
| Colville | N. America | (68, -154) | 57544 | 31.9 | 2.66 |

## Appendix B: SPEI computation

SPEI is computed by aggregating and normalizing a simple climatic water balance,

$$D_i = P_i - PET_i, \tag{B1}$$

where $P_i$ is the precipitation in time step $i$, $PET_i$ is the potential evapotranspiration in the same time step, and $D_i$ is their
difference. We take precipitation $P_i$ directly from the output of each GCM that we analyze. We estimate $PET_i$ with the Penman-
Montieth method, following Allen et al. (1998). To calculate PET requires surface temperature, surface pressure, surface
specific humidity, and surface net radiation from the GCM. Surface wind is set to be constant, as PET has been shown to be
insensitive to the inclusion of surface wind from GCMs (Cook et al., 2014). We account for time-varying stomatal conductance
of vegetation cover following Yang et al. (2019). $P$ and $PET$ are computed as basin-total quantities by extracting the necessary
variables for individual GCM grid cells and performing an area-weighted sum over all grid cells that intersect each basin (with
basin boundaries defined by Global Runoff Data Centre, 2007). In the version of SPEI that accounts for glacial runoff, we
modify the basin total precipitation term $P$ to include glacial runoff $R$, as shown in Equation 1.

The climatic water balance $D$, integrated to various timescales, is then standardized over the period 1900-1979, by ranking
each integration period's moisture balance in a nonparametric distribution of the values from 1900-1979 for the same integra-
tion period (after Farahmand and AghaKouchak, 2015). The glacier model output is available only from 1980 onward and is

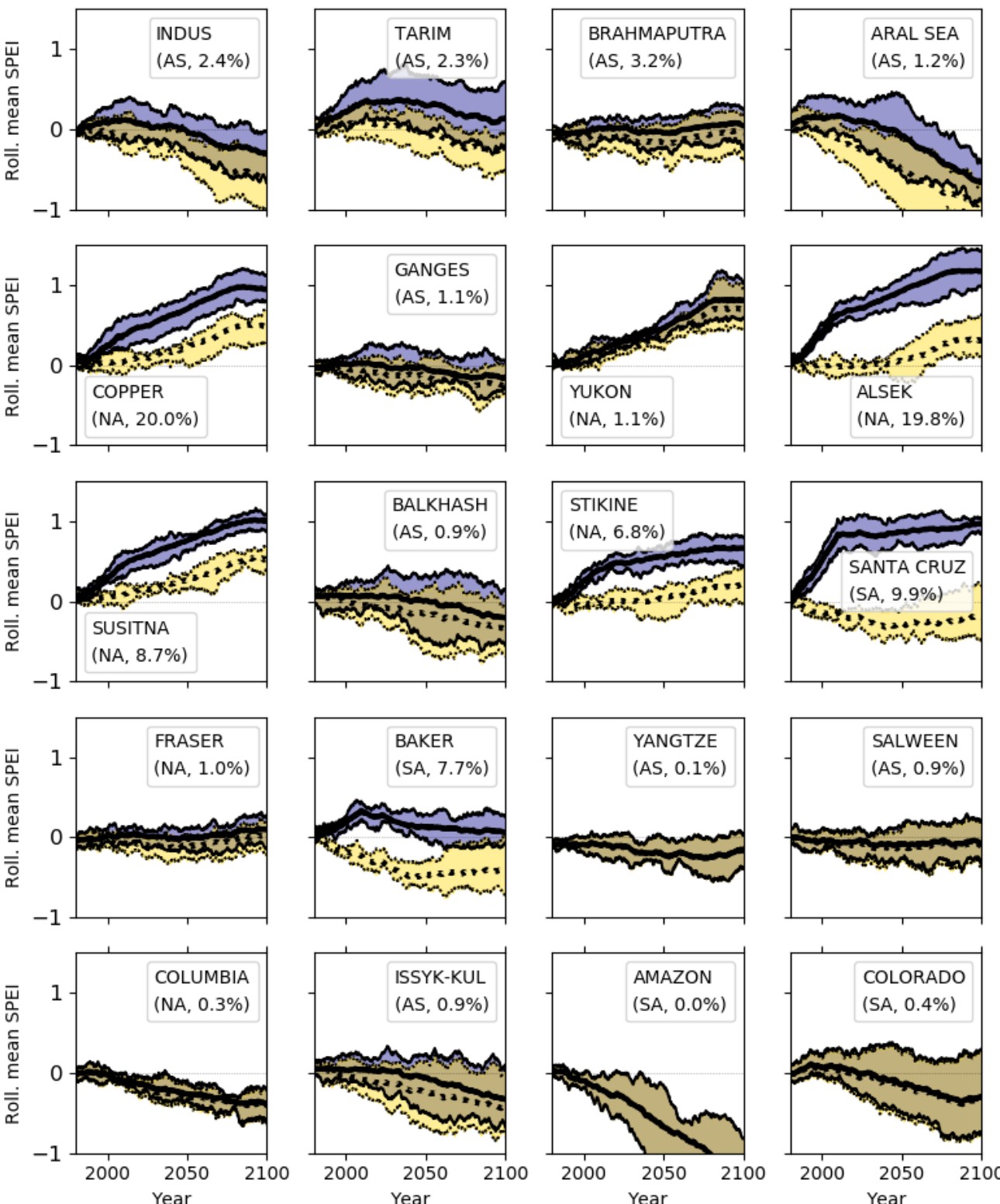

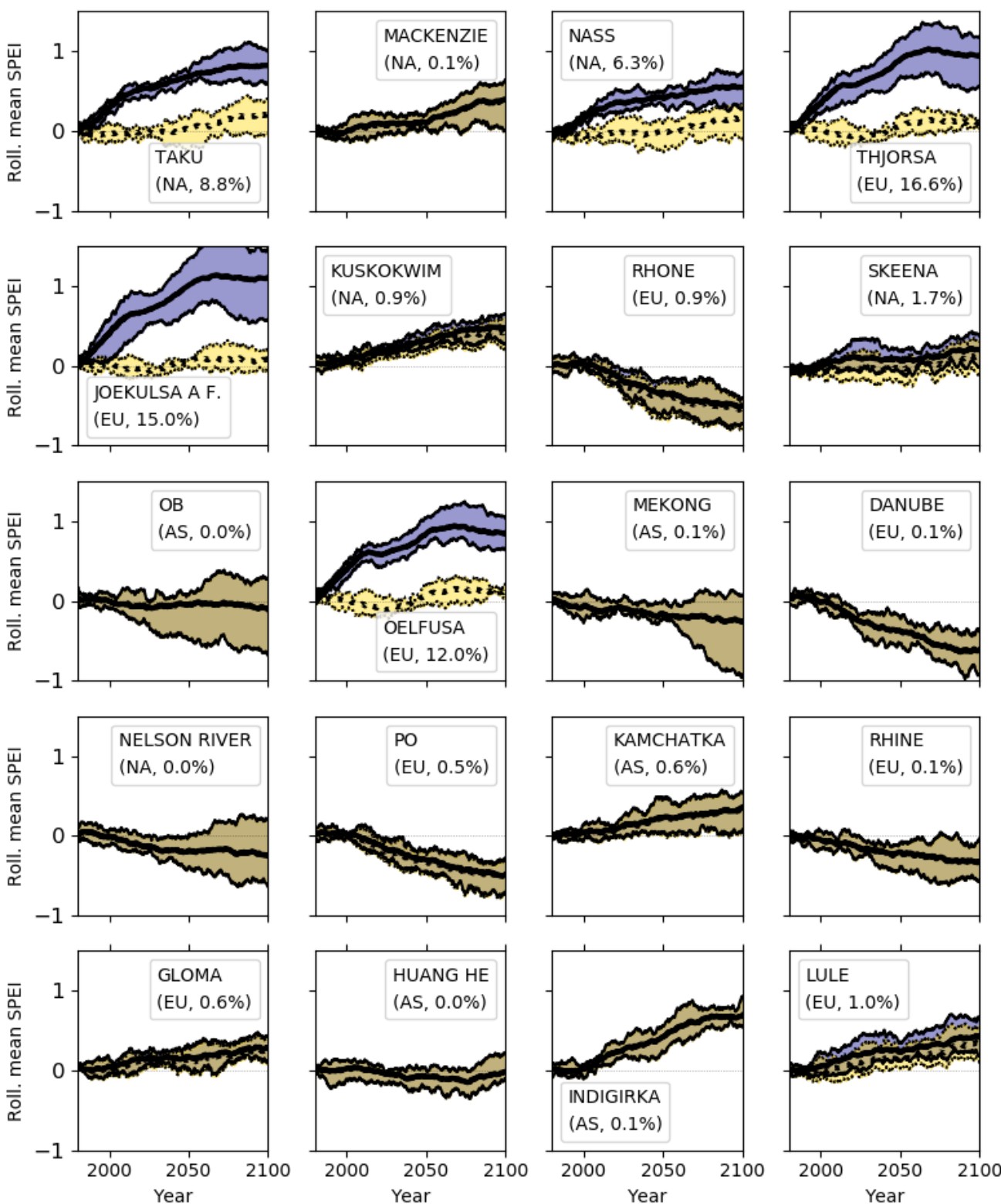

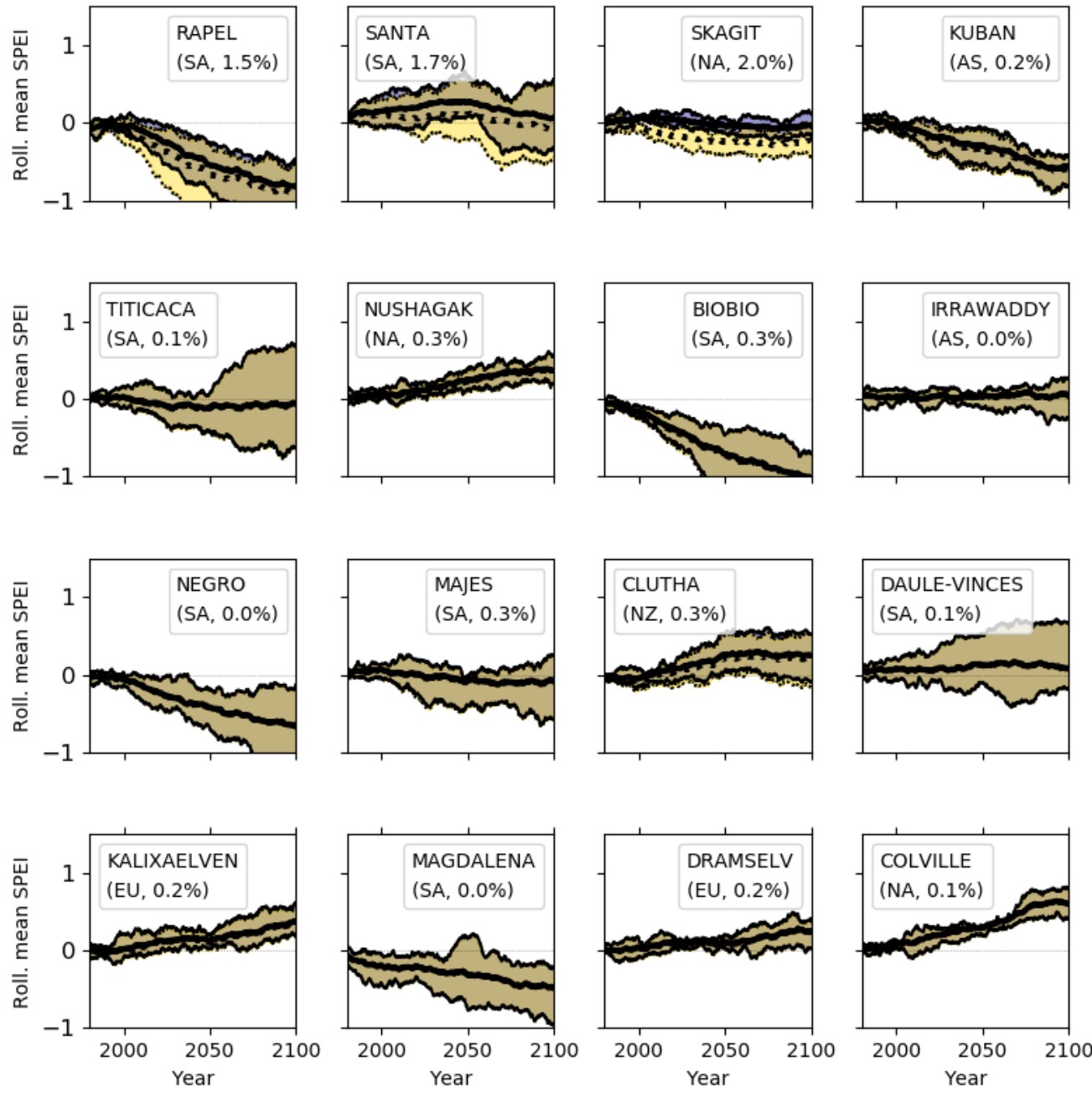

**Figure A1.** 30-year running ensemble mean (lines) and interquartile range (shaded region) of SPEI$_{3mo}$ with glacial runoff (blue shades) and without (yellow shades), for the RCP 4.5 scenario, similar to main text Figure 1. Basins are presented in descending order of total glacier area. Unlike Figure 1, all panels here share a common y-axis scale for ease of comparison. As a result, smaller-magnitude shifts are less apparent here. Basins are labelled by name, regional abbreviation, and percent glacial cover. Abbreviations are 'AS' for Asia, 'EU' for Europe, 'NA' for North America, 'NZ' for New Zealand, and 'SA' for South America.

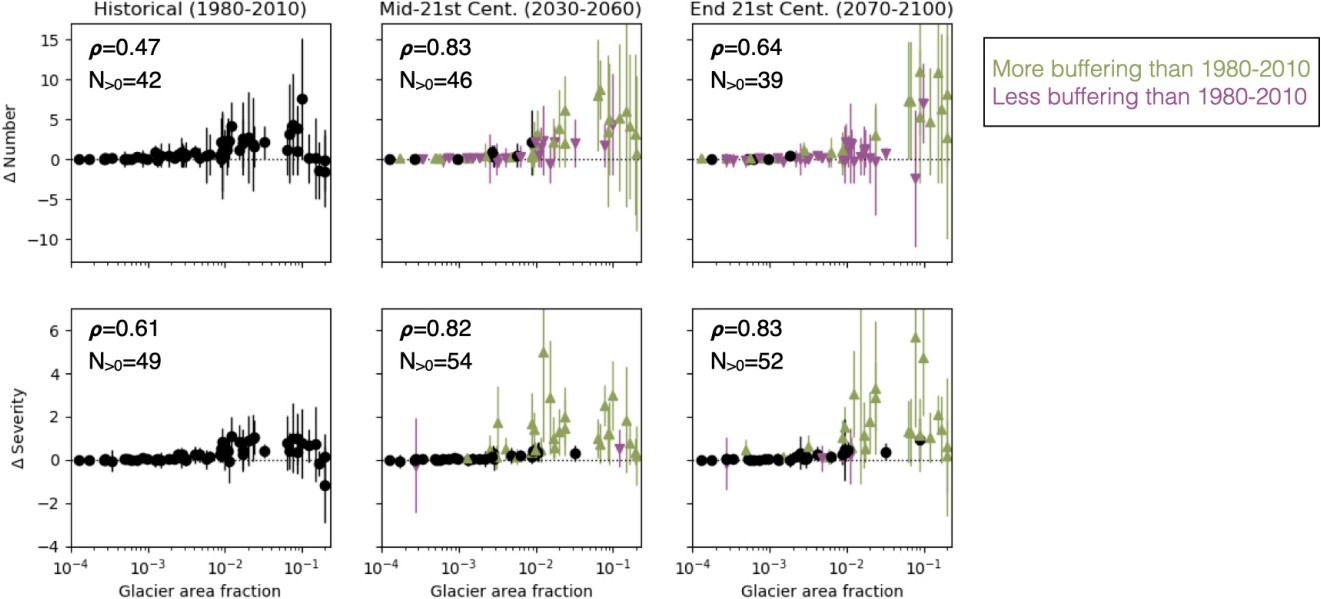

**Figure A2.** Glacial buffering of number of droughts (top row) and drought severity (bottom row), for the periods 1980-2010 (left), 2030-2060 (center), and 2070-2100 (right), as in main text Figure 3 but for RCP 8.5. Differences are given as the statistic on $SPEI_N$ - $SPEI_G$, so a positive $y$-axis means positive buffering, i.e. there are more, and/or more severe, droughts in the without-glacial runoff series. X-axes are glacier area fraction $A_G/A$ presented on a logarithmic scale. Markers indicate the ensemble mean for each basin and whiskers indicate range among single-GCM results. In the second and third columns, color indicates change in ensemble mean buffering versus the historical period: green upward triangles for increased buffering, black dots for no change (tolerance of $\pm 0.1$), and magenta downward triangles for decreased buffering. Annotations in the corner of each panel list Spearman's rank correlation coefficient $\rho$ between basin glaciated area and the statistic shown in that panel, as well as the number of basins for which multi-model mean buffering is positive, $N_{>0}$.

thus not included in the standardization set. Although it would be desirable to standardize glacial SPEI with a standardization set that included glacial runoff, the available data limits the possible standardization interval to an interval that might not capture the spectrum of relevant climate variability. Therefore, the standardization sets for $SPEI_G$ and $SPEI_N$ are identical. This methodological choice may tend to overestimate the glacial effect on baseline SPEI (Figures 1-2), but will have little impact

on the change in drought buffering over the 21st century computed with reference to 1980-2010 (Figures 3-4).

## B1    Sensitivity to integration timescale

SPEI includes a user-selected timescale of integration, which can be adjusted to study different types of drought and different parts of the hydroclimate system. Short timescales relate to availability of water as soil moisture and headwater river discharge, while longer timescales relate to reservoir storage, downstream water discharge, and changes in groundwater storage (Vicente-

Serrano et al., 2009).

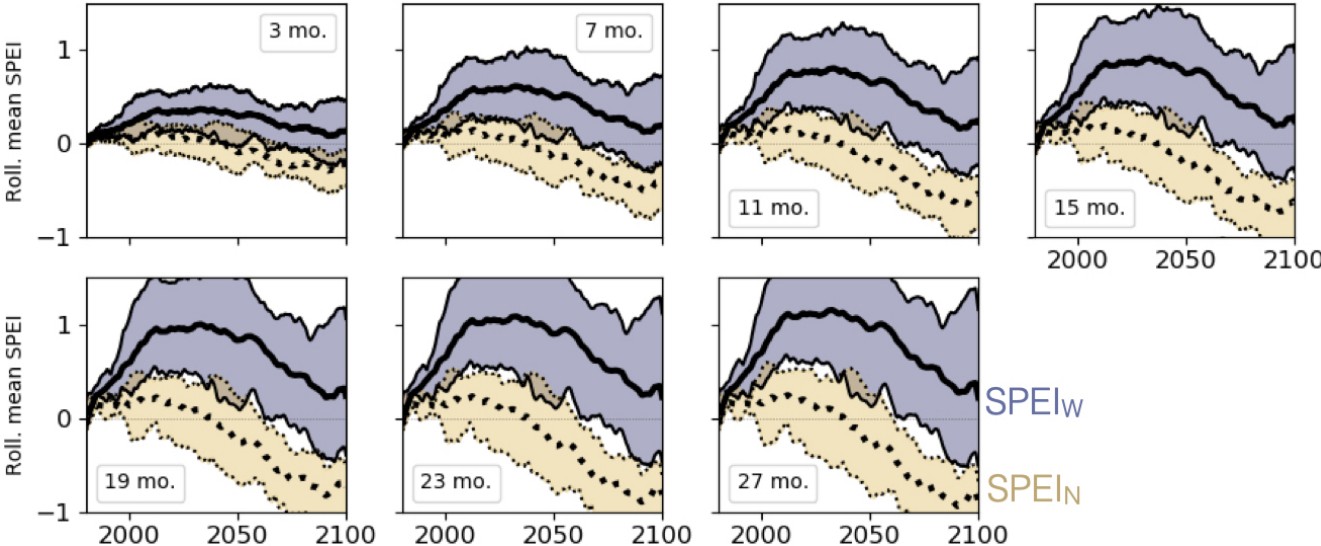

**Figure B1.** Comparison of glacial effect on SPEI over the 21st century when SPEI is computed with integration timescales ranging from 3 to 27 months, shown for the Tarim basin as an example. The upper left panel shows the 3-month integration timescale presented in the main text. Color scheme and axes are consistent with Figure A1.

We computed SPEI at a range of integration timescales and found that while there were quantitative differences in the results, for example the magnitude of the baseline effect on SPEI as shown in Figure 2, the results were qualitatively similar across timescales. One example is below; results for all basins and timescales are available on our public repository.

Figure B1 shows the glacial effect on mean SPEI in the Tarim basin (compare with Figure 2b), with SPEI computed at seven different timescales of integration. The qualitative patterns of the glacial effect on SPEI is similar across integration timescales: the ensemble mean of $SPEI_G$ shows an initial increase in water availability compared with 1980-2012, with subsequent decline but a nevertheless positive ensemble mean SPEI through the end of the century. Rolling ensemble mean $SPEI_G > SPEI_N$ for every integration timescale, though the interquartile range is larger for longer integration times, suggesting greater structural uncertainty on these timescales. The magnitude of the difference $\Delta SPEI = SPEI_G - SPEI_N$, computed for each GCM, ranges from 0.1 to 2 SPEI units at different timescales. Although the smallest-magnitude effect appears at the shortest timescale of integration in the Tarim basin, there is no monotonic relationship between $\Delta SPEI$ magnitude and integration timescale. That is, the magnitude of the effect we analyse does not scale linearly with the SPEI integration timescale.

## B2   Accounting for glacial runoff

We account for glacial runoff in each basin during the period 1980-2100 using the runoff simulations of Huss and Hock (2018). Their model is forced by monthly near-surface air temperature and precipitation from global climate reanalysis (Dee et al., 2011) and CMIP5 GCM projections (Taylor et al., 2011), downscaled to each individual glacier. The initial area of each glacier is defined as the "glacier catchment" for the duration of the simulation. That is, the portion of a basin within a glacier

catchment does not change over time, even as the area of the glacier itself does change. Runoff is simulated at the individual glacier level and includes all water exiting the catchment, both melted snow and ice as well as rain falling within the catchment boundary. These monthly glacier runoff totals are then aggregated to the basin scale.

In the Huss and Hock (2018) glacial model output, some portion of the GCM-derived precipitation falling within a basin is also counted within the basin glacial runoff. To avoid double-counting precipitation in our $\text{SPEI}_G$ moisture source term, we scale GCM-derived precipitation by each basin's unglaciated area (Equation 1) and add it to glacial runoff. PET is then subtracted from this sum, which is equivalent to assuming that both precipitation falling in the unglaciated part of the basin and glacial runoff from the glaciated part of the basin are encountering atmospheric demand for moisture.

### B2.1 Non-glacial effects deriving from our processing

There are two model implementation details that may tend to overestimate the moisture source term in our Equation 1. First, as described above, we have scaled basin total precipitation by each basin's unglaciated area and then added the glacial runoff, rather than scaling down the precipitation in the specific grid cells where glaciers are located. If glaciers are concentrated in comparatively wet grid cells, as in some Central Asian basins (Immerzeel et al., 2020, Extended Data Figure 3a, with thanks to reviewer Sarah Hanus), our method will tend to overestimate total precipitation in Equation 1. Second, the model of Huss and Hock (2018) includes a basin-dependent precipitation scaling factor to account for high-elevation precipitation that is underestimated in GCM output (Huss and Hock, 2015). The basin glacial runoff $R$ that we use in Equation 1 therefore depends on the locally scaled precipitation, i.e. model implementation, as well as the glacier dynamics that are our focus.

To examine the implications of these two non-glacial effects for our drought-buffering results, we computed a version of SPEI with an enhanced moisture source and no glacial runoff included:

$$\tilde{P}_{\text{test}} = \frac{A - A_g}{A} P + (\frac{A_g}{A} * c_{\text{prec}} * P), \tag{B2}$$

where $c_{\text{prec}}$ is the precipitation enhancement factor. Because $c_{\text{prec}}$ is calibrated per glacier in Huss and Hock (2015), and we work with basin-aggregated data, a rigorous analysis of the effect of precipitation scaling on SPEI in our global results is beyond the scope of the present study. Instead, we examine the general effects of moisture overestimation for a few example basins using $c_{\text{prec}}$=1.5, a default value taken from Huss and Hock (2015). We compared this general enhanced-moisture-source $\text{SPEI}_{PS}$ against $\text{SPEI}_N$ and $\text{SPEI}_G$ for some example basins, as shown in Figure B2. For the high-precipitation, heavily-glaciated Copper basin, the enhanced-moisture-source $\text{SPEI}_{PS}$ ensemble overlaps with the glacial $\text{SPEI}_G$, indicating that the drought buffering we find there may include non-glacial effects. For the other example basins, the enhanced-moisture-source $\text{SPEI}_{PS}$ has more overlap with $\text{SPEI}_N$ than with $\text{SPEI}_G$, indicating that the drought buffering in those basins is more likely a direct result of glacier dynamics.

We also computed the differences in drought frequency and severity identified in $\text{SPEI}_{PS}$ series versus $\text{SPEI}_N$ series, reflecting the the drought buffering that could be a result of non-glacial overestimation in moisture source. Table B1 shows a comparison of the drought buffering statistics as originally shown in Figure 3 and those computed for the precipitation-scaling test, for two example basins. The Copper and Indus are chosen as examples of high-precipitation and high-potential-evapotranspiration

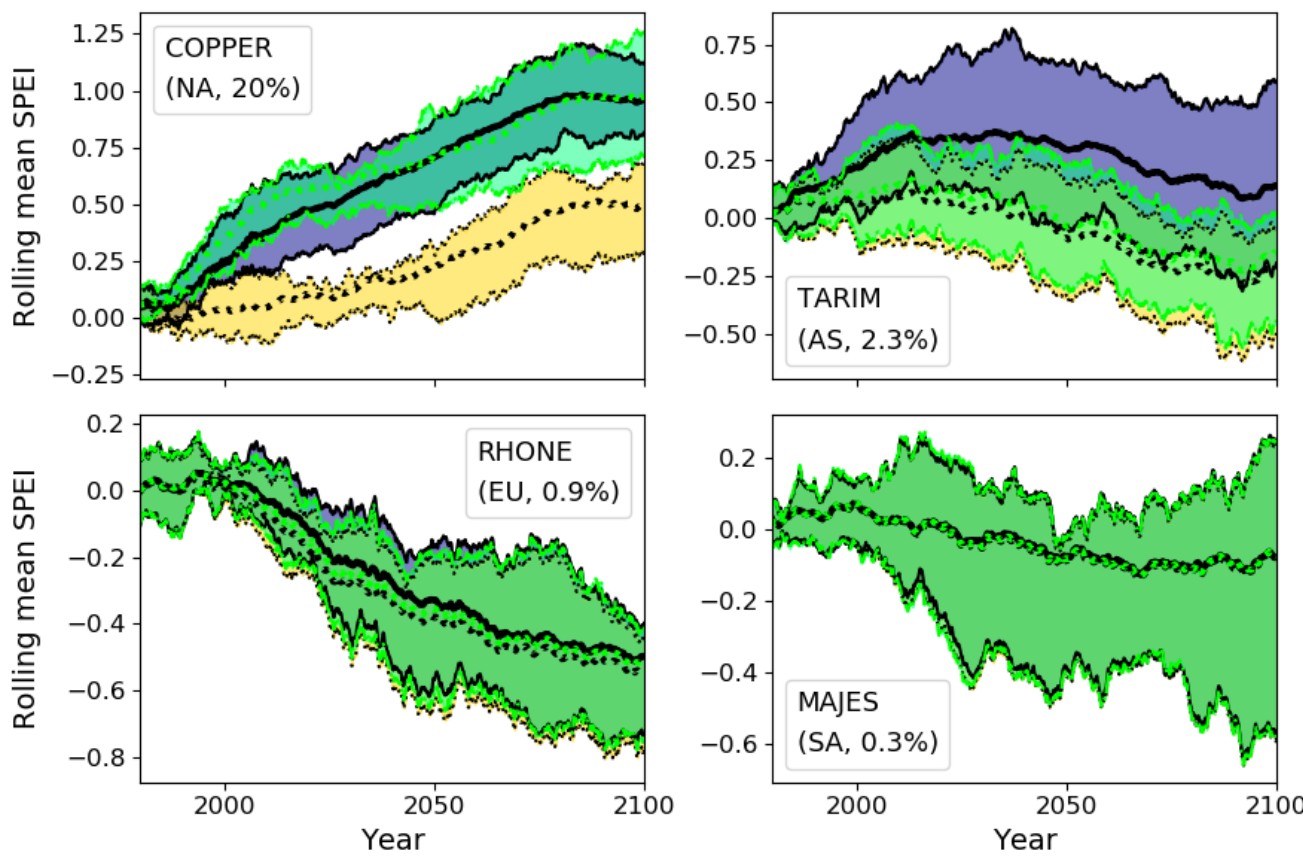

**Figure B2.** 30-year running ensemble mean and interquartile range of $\text{SPEI}_{3mo}$ unmodified (yellow shades), with glacial runoff (blue shades), and with precipitation scaled up as described in B2.1 (green shades, bright outlines) for the RCP 4.5 scenario in four example basins. As in Figure 1, each panel is annotated with basin name, region abbreviation, and percent glacier coverage by area. 'NA' indicates North America, 'AS' Asia, 'EU' Europe, and 'SA' South America.

basins, respectively, where a scaling-up of precipitation would be expected to have a strong effect on SPEI. We find that the patterns of glacial drought buffering in $\text{SPEI}_G$ are distinct from the hypothetical buffering shown by $\text{SPEI}_{PS}$ for both basins. Both the magnitude of buffering and the trends in each buffering metric over time are different between the glacial case and the precipitation-scaling case. We conclude that non-glacial overestimation of the moisture source in SPEI is unlikely to be a major factor in the drought buffering results we present above.

### B3 Identifying droughts

We identify droughts in the single-GCM time series of SPEI as described in Section 2.3. Figure B3 provides an illustration for a time series of SPEI for the historical period 1980-2010, computed for the Tarim basin from the GCM CCSM4, without

| Basin | Statistic | Case | Historical | Mid-century | End-century |
|---|---|---|---|---|---|
| INDUS | Δ **Number** | Glacial runoff | 1.75 | 2.75 | -0.25 |
| | | *Precip. scaling* | *0.25* | *0.25* | *0.88* |
| | Δ **Severity** | Glacial runoff | 1.02 | 1.22 | 3.22 |
| | | *Precip. scaling* | *0.16* | *0.15* | *0.53* |
| COPPER | Δ **Number** | Glacial runoff | -1.25 | -0.5 | 1.13 |
| | | *Precip. scaling* | *7.5* | *6.13* | *4.88* |
| | Δ **Severity** | Glacial runoff | -1.16 | -0.03 | 0.10 |
| | | *Precip. scaling* | *1.31* | *0.97* | *1.01* |

**Table B1.** Comparison of drought buffering statistics computed with glacial runoff (SPEI$_N$-SPEI$_G$, as in main text Figure 3) versus buffering from enhanced precipitation alone (SPEI$_N$ - SPEI$_{PS}$, as described in B2.1). Statistics shown are multi-GCM means. Note that the Copper basin is heavily glaciated and has high precipitation; see Section B4 for a discussion of negative glacial buffering in such settings.

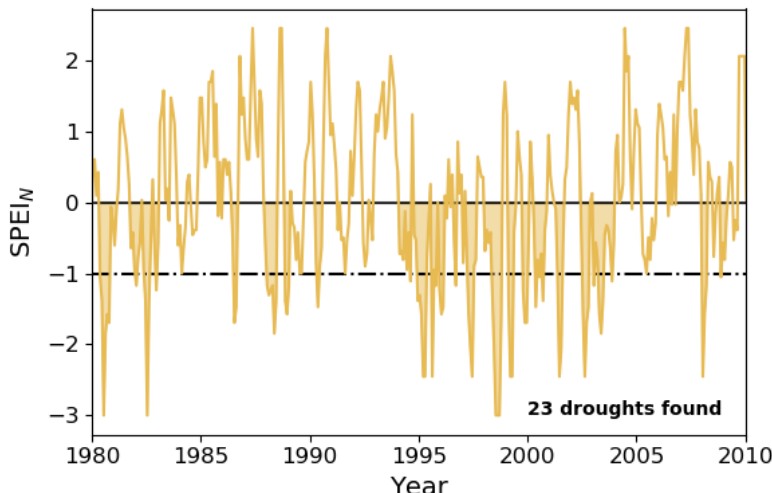

**Figure B3.** SPEI$_N$ time series for the Tarim basin, 1980-2010, with droughts shaded. The accumulated SPEI deficit we analyse as "severity" for each drought can be visualized here as the sum of monthly SPEI values in a continuous shaded area.

accounting for glacial runoff. The shaded intervals indicate droughts as defined in Section 2.3. The accumulated SPEI deficit
(i.e. the integral of the SPEI curve) during each drought is its "severity". We have annotated the figure to indicate that our method counted 23 droughts during the period of this example.

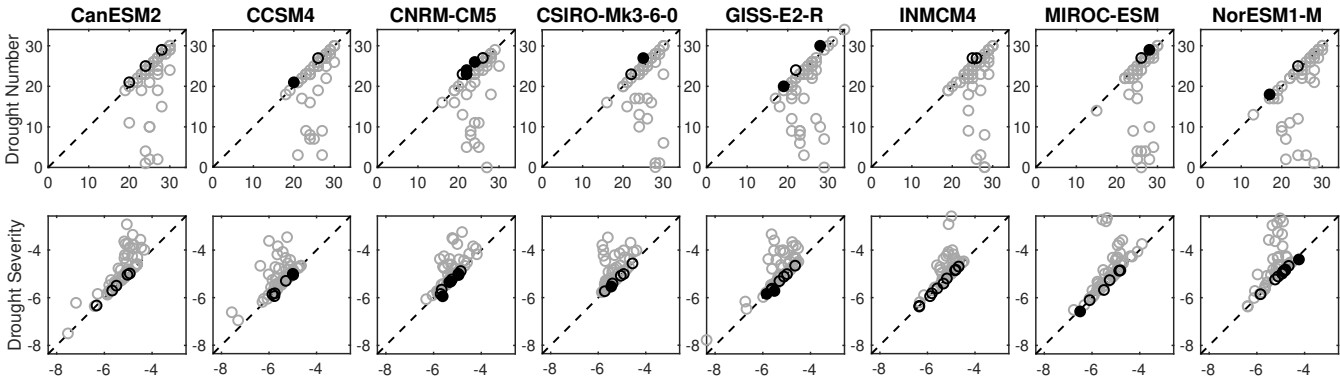

**Figure B4.** Scatter plot showing number of droughts (top row) and drought severity (bottom row) for $SPEI_G$ on the y-axis and $SPEI_N$ on the x-axis. Each column is a single GCM and each dot represents a single basin. Black outlined points indicate basins with negative buffering for the metric and GCM provided. Filled black points indicate basins with negative buffering across both metrics for the given GCM (what we describe as "true" negative buffering).

## B4 Negative buffering

In certain cases, there is an apparent negative buffering effect (negative y-axis values in Figure 3). That is, we identify more droughts and/or more severe droughts in the $SPEI_G$ time series than in the $SPEI_N$ series. There is a physical reason to expect some negative buffering as well as a numerical reason why it might be found in our analysis; we outline both in this section.

### B4.1 Incidence of "true" negative buffering

We define "true" negative buffering as a case for which the $SPEI_G$ series has both more, and more severe droughts than $SPEI_N$. The first row of Figure B4 shows mean drought number with glacial runoff on the y axis and without glacial runoff on the x axis for the historical period–each subplot is a model and each dot is one basin. If a dot falls below the one to one line then there is positive buffering for that basin/model combination, while a dot above the 1:1 line indicates negative buffering. The second row shows the same comparison for drought severity (accumulated SPEI deficit); note that the axis orientation is reversed for drought severity and so negative buffering appears below the 1:1 line. In total, there are only 9 basin/model combinations that have true negative buffering, and the magnitude of any negative buffering is very small (nothing is far away from the one to one line in the negative buffering direction in Figure B4).

There is not one model or one basin that produces consistent true negative buffering. However, all 9 basin/model combinations that show negative buffering in the historical period come from basins 1 to 8 ranked by basin glacier area. We interpret that only heavily glaciated basins can have true negative buffering as identified in our method. Figure B5 shows the climatology for the historical period of (going left to right) PET, P, glacial runoff, P-PET, P-PET+glacial runoff for the 9 basin/model combinations with true negative buffering. On the bottom is a random sample of basin/model combinations that do not have true negative buffering taken from basins 1 to 8 ranked by basin glacial area.

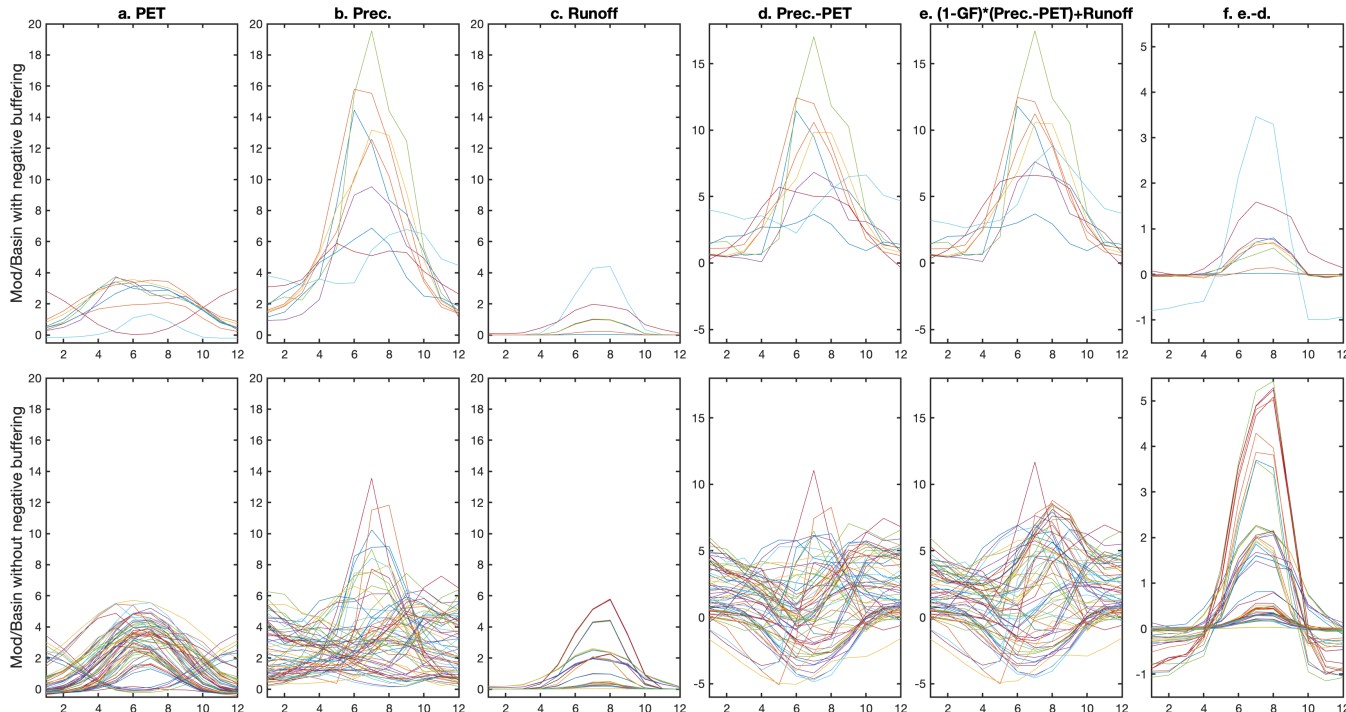

**Figure B5.** Climatology over the historical period (1980-2010) for five quantities in the moisture balance: a. Potential evapotranspiration (PET), b. Precipitation (P), c. Glacial runoff, d. Moisture balance to be standardized in $\text{SPEI}_N$ ($P-PET$), e. Moisture balance to be standardized in $\text{SPEI}_G$ accounting for glacial runoff, f. Difference between balance terms. Top row: quantities corresponding to the 9 basin/model combinations that show true negative buffering over the historical period. Bottom row: Quantities from a random sample of basin/model combinations among the top 8 basins by glacial area.

The only notable difference between the two sample sets is in the magnitude of precipitation. The basin/model combinations with true negative buffering have relatively large precipitation magnitudes and ubiquitously positive P-PET. In heavily glaciated basins ($A_g/A$ large) with periods of low runoff, there may arise an imbalance when $R < \frac{A_g}{A}(P - Pet)$, such that the modified water balance in Equation 1 is lower than the original.

The lack of other major differences between the two sample sets, or a single model susceptible to true negative buffering, suggests that the true negative buffering is random and emerges because of the variability over the historical period.

### B4.2    Effect of shift in seasonality

Glacial runoff has a seasonal signature distinct from that of precipitation-driven runoff, generally peaking later in the warm season when rains have already subsided (Hagg et al., 2013; Huss and Hock, 2018; Barandun et al., 2020). As a result, in

basins where glacial runoff is a substantial fraction of total runoff, accounting for glacial runoff can shift the moisture balance from month to month. Our SPEI computation, which standardizes by ranking each integration period's moisture balance in a

nonparametric distribution of historic values for the same integration period, may then produce a $\mathrm{SPEI}_G$ value lower than the $\mathrm{SPEI}_N$ value for a short period. Those short intervals of lower $\mathrm{SPEI}_G$ may then be identified as droughts that are not present in $\mathrm{SPEI}_N$. However, $\mathrm{SPEI}_G$ is not systematically less than $\mathrm{SPEI}_N$ for any basin, and in general the moisture balance "deficit" identified in one month due to this shift in seasonality may be expected to be recovered in a subsequent month.

## Appendix C: Standardized Runoff Indicator versus SPEI

Among the many drought indicators in current use (see World Meteorological Organization and Global Water Partnership, 2016), we chose to analyse SPEI (Section 2 and Appendix B). The Standardized Runoff Indicator (SRI) is another drought indicator that could be modified to include glacier runoff input. Several GCMs report runoff directly as a model output, which simplifies the analysis of SRI as contrasted with SPEI. However, availability and quality of GCM-derived runoff data is limited. Where data was available, we computed the correlation between SPEI and SRI for each GCM, for every basin, over three 50-year time periods. We found that the median correlation was approximately $0.5$. Given the generally positive correlation of SPEI with SRI, and the greater data availability for SPEI ($\sim 36\%$ more GCM-basin pairs), we proceeded with SPEI for our analysis.

*Author contributions.* EHU and SC designed the study, with input from JM. SC computed SPEI and analyzed global climate model output. EHU performed statistical analysis and produced most figures. EHU and SC wrote the first draft of the manuscript. JM contributed subsequent analysis, including k-means clustering for Figure 4, and helped to edit the final version. JM publishes with the permission of the Executive Director, British Geological Survey (UKRI). All authors have reviewed the manuscript and approved its conclusions.

*Competing interests.* The authors have declared that no competing interests are present.

*Acknowledgements.* Monthly glacial water runoff model results were kindly provided by Dr. Matthias Huss. The authors thank four anonymous reviewers for their constructive comments on previous versions of the manuscript. Reviewers Sarah Hanus and Caroline Clason offered extremely helpful comments on this manuscript, which led to substantial improvement in the work. This manuscript is SOEST publication number [XXXXX - *number will be provided if accepted and must be added to final version*].

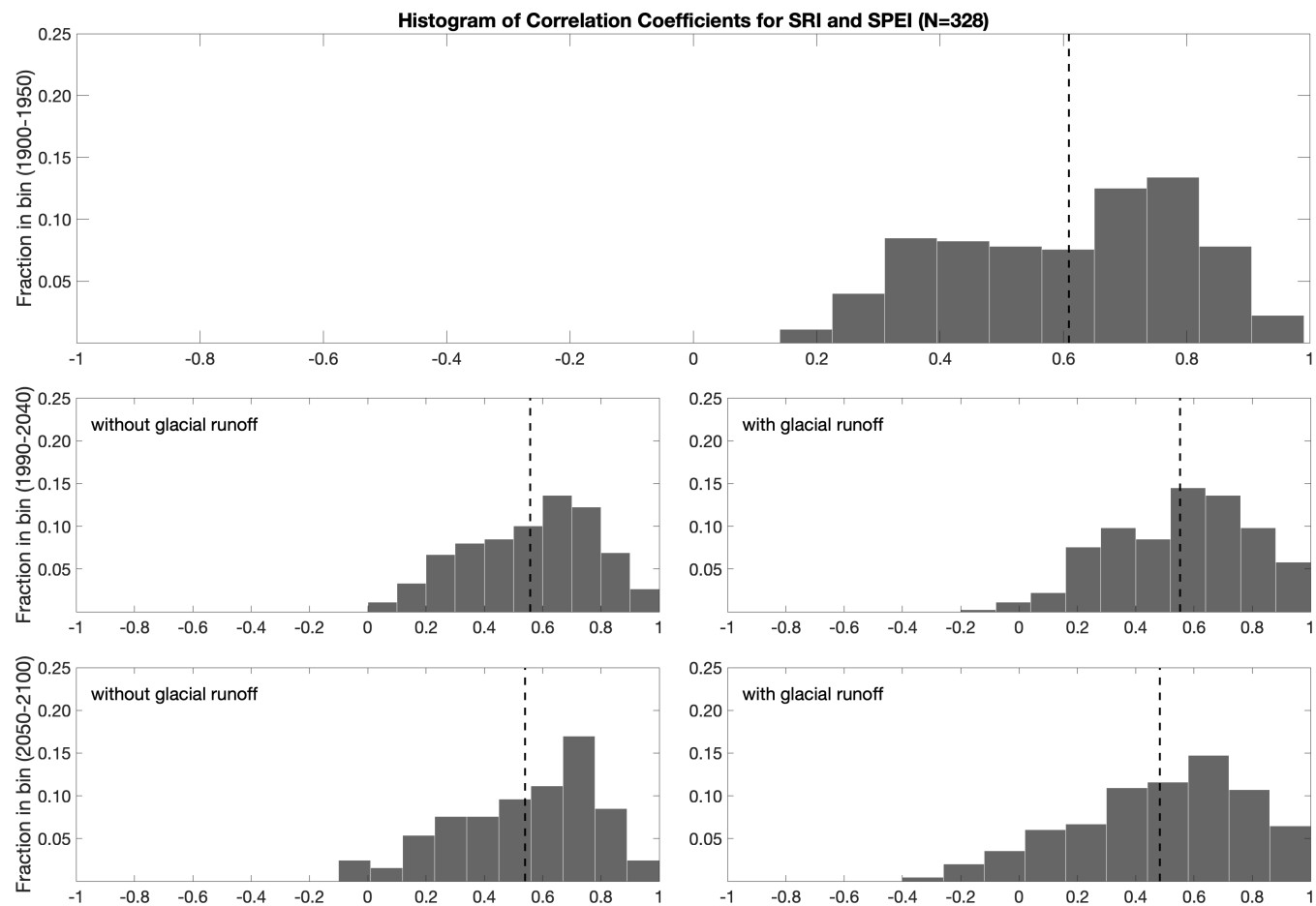

**Figure C1.** Histograms of correlation coefficients between Standardized Runoff Indicator (SRI) and SPEI, computed over three time periods. Top panel: 1900-1950, when there is no glacier model input available. Middle row: 1990-2040, correlating SRI with $SPEI_N$ (left) and $SPEI_G$ (right). Bottom row: 2050-2100, correlating SRI with $SPEI_N$ (left) and $SPEI_G$ (right). Dashed lines indicate median correlation values.

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
