# Peer review of "Glacial runoff buffers droughts through the 21st century"

_Earth System Dynamics, 2021_

## Referee Comment (RC3)

**Paper Review: Glacial runoff buffers droughts through the 21st century (Ultee et al.)**

**Overview**

This paper investigates drought buffering by glacier runoff in the 21$^{st}$ century in 56 major river basins worldwide. Glacier runoff can be an important buffer to interannual variability in runoff. However, dynamic glacier runoffs are not included in GCMs and thus also neglected when calculating drought indices (e.g. SPEI). The authors modify the SPEI to consider glacier runoff as part of water accumulation. Using the SPEI, they calculate number and severity of drought events from 1980-2100 using 8 GCMS and RCP 4.5 and RCP 8.5. They compare differences of the modified SPEI including glacier runoff to normal SPEI. Moreover, they perform a correlation and clustering analysis to identify the explanatory variables for drought buffering by glacier runoff. The results suggest an increased water supply and a buffering of drought frequency and severity when accounting for glacier runoff. A large glaciated fraction and low historical precipitation lead to high drought buffering by glaciers as identified by the clustering analysis.

**General comments**

This is a very well-written paper focusing on the relevance of glacier runoff for future hydrological droughts worldwide. I enjoyed reading it because it is well structured, concisely written with self-explanatory headers, and it provides a novel analysis of importance of glaciers. I also appreciate the code and data availability including a Jupyter Notebook to reproduce the results.

**Specific Comments**

**Methodological Limitations / Uncertainties**

- In line 115 you mentioned an important assumption of your approach and claim that you will address this simplification below. However, I did not find where this assumption was further addressed in the discussion or elsewhere. Precipitation in mountain areas (where glaciers are located) tend to be larger than in the rest of the basin due to orographic precipitation, especially in basins, which are considerable arid further downstream (e.g. Amu Darya) (Immerzeel et al., 2020, Extended Data Fig.3a). Assuming an equal distribution of precipitation when calculating the modified moisture source term leads most likely to an overestimation of total precipitation in the basin. Thus, the moisture source term in SPEI will be increased, however this increase is an artefact of the method used and not due to including glacier melt. I miss a discussion about this limitation and how it might have influenced the results and conclusions. I would expect that the influence is larger in basins that are considerable drier in the lowlands than in the mountains.

    Below is a simplified example to illustrate my point using arbitrary values:

    Precipitation in Mountains (20% of catchment): 3000mm

    Precipitation in Lowlands (80% of catchment) 500mm

    Glacier Area 4%

    Total precipitation: 0.2*3000 + 0.8*500 = 1000 mm

    In the modification of your precipitation input this would result in a total precipitation of

    1000*0.96 + 0.04*3000= 1080mm

- In the model of Huss and Hock (2015) which is used to generate the glacier runoff in Huss and Hock (2018) a precipitation correction factor and a precipitation lapse rate is applied to the input precipitation data under the assumption that precipitation in high elevations is likely underestimated in data sets. The precipitation correction factor is used as calibration parameter for the glacier model. Therefore, precipitation as output from the glacier model is different from the original precipitation data set. So, the runoff from glacier area in form of snowmelt and rain will be higher than the precipitation of the original GCM data on this area. This also leads to an increase in the moisture source term of SPEI that does not directly stem from glacier melt itself. Although glacier area is shrinking, the precipitation input of the glacier model is still used for the initial glacier area. I would appreciate a discussion about the possible limitation: How does the usage of total glacier runoff data fixed to the initial glacier area instead of using only glacier melt data impact your results? As an idea, you could increase the moisture source term of $SPEI_N$ by increasing the precipitation over the glacier area by a default factor (e.g. 1.5 as this is the default factor for precipitation correction in GloGEM (Huss and Hock, 2015)) and look if this impacts your results (Fig. 3)

$$P_{SPEI_N} = \frac{(A - A_G)}{A} \cdot P + \frac{A_G}{A} \cdot 1.5 \cdot P$$

- In general, I missed a bit more in depth discussion about limitations and uncertainties of the study. It was touched upon (l.249- 257) but this is only related to possible feedbacks of glacier retreat that are not captured in the GCMs, but I think there would be more uncertainties to discuss, for example how informative it is to average hydro-climatic variables over the whole basins, and to discuss the possible limitations stated above.

-

**Basins without drought buffering of glaciers?**

- Figure 3: The figure contains a lot of data points. Therefore, it is difficult to grasp, how many basins experience more/less/equal buffering in future and how many basins do not experience a buffering at all in the past and the future (black dots on the zero line). I would suggest to either add these numbers in the figure or in the text describing the figure. In the abstract you claim that accounting for glacier runoff tends to increase drought buffering even in basins with little glacier cover. Looking at Figure 3, I see also a lot of basins without difference especially for small glaciers (black dots close to the zero line). How do you explain this?

- How do you explain that there are quite a lot of basins where you do not see drought buffering in 1980-2010 in Figure 3, although you use the same standardization set for $SPEI_w$ and $SPEI_N$ as explained in l.305-312. I would argue that this should lead to higher $SPEI_w$ values than $SPEI_N$ values in the past and thus maybe to a reduced number and severity of droughts for $SPEI_w$ compared to $SPEI_N$.

**Minor Comments**

Abstract

- L.4 I would specify "hydrological drought conditions".
- L.7 can you quantify more what "relatively little" glacier cover means?

- I would appreciate some more quantification of the results, e.g. for how many basins did the inclusion of glacier runoff buffer droughts?

Methodology

- L.95: A list of used GCMs in the appendix would be useful for the reader (now you have to search for it until you find it in Figure B3)
- L 137: A short explanation of the definition of aridity index you use would be helpful. I assume you use ratio of precipitation to evaporation such that high values mean wetter conditions? (compare l.201, l.235)

Results

- Figure 1 and Figure A1: I think adding the percentage of glacier coverage next to the name of each basin would give the reader a better overview of the glaciation of the basin and make it easier to interpret the results of the figure
- In Figure 3, Table 1 and Figure 4 I miss the description of whether RCP 4.5 or RCP 8.5 was used.
- Figure 3: no units for the severity are given as axis labels.
- Table 1: AI: abbreviation not mentioned before

Appendix

- L.368: "The only notable difference between the two sample sets is in the magnitude of glacial runoff." This statement is not clear to me, I do not see it in Figure B4. I rather see differences in precipitation between the upper and lower row.
-

**Techincal corrections**

- L. 339 SPEI$_G$ should be SPEI$_W$

*References*

Huss, M., & Hock, R. (2015). A new model for global glacier change and sea-level rise. *Frontiers in Earth Science*, *3*, 54.

Huss, M., & Hock, R. (2018). Global-scale hydrological response to future glacier mass loss. *Nature Climate Change*, *8*(2), 135-140.

Immerzeel, W. W., Lutz, A. F., Andrade, M., Bahl, A., Biemans, H., Bolch, T., ... & Baillie, J. E. M. (2020). Importance and vulnerability of the world's water towers. *Nature*, *577*(7790), 364-369.

---

## Author Comment (AC2)

**Author Response to Reviews of**

**Glacial runoff buffers droughts through the 21st century**

L. Ultee, S. Coats, J. Mackay
*Earth System Dynamics Discussions*
* * *
RC: *Reviewer Comment*,    AR: *Author Response*,    □ Manuscript text

We thank all three reviewers for their thoughtful and extremely prompt reviews. Each reviewer offered suggestions that helped to improve the manuscript, and we enjoyed considering their ideas. We respond to each suggestion directly below, where necessary quoting excerpts of revised text. We are prepared to submit a full revised manuscript when appropriate.

**1.  Anonymous Reviewer 1**

**1.1.  General comments**

RC: *In the context of future climate change, to quantitatively analyze the effects of glacial runoff on droughts is of great significance. The topic of this manuscript is important; however, this manuscript, in its present form, still contains several weaknesses. My major concerns are as follows:*

*First, there is a lack of recent references (NONE published in 2021). This manuscript will be less persuasive without a comprehensive literature review. I would suggest the authors review and cite more relevant papers.*

AR: *Please see our previous Author Comment regarding the comprehensiveness of the literature review. In the meantime, we have added the following two 2021 citations to the Discussion:*

- *Shaw et al 2021, Glacier albedo reduction and drought effects in the extratropical Andes, 1986–2020,* Journal of Glaciology

- *Hugonnet et al 2021, Accelerated global glacier mass loss in the early twenty-first century,* Nature

RC: *Second, the authors use SPEI (rather than SPI) as an indicator of drought considering runoff. Although they have stated in Appendix C that "SRI has a moderate to strong positive correlation with SPEI", the median correlation is only approximately 0.5, which is not high. I would suggest the authors further discuss the influences of using SPEI on the results. Otherwise, the conclusions are not reliable enough.*

AR: *We outline our reasons for choosing SPEI in the main text. Regarding the correlation of SPEI with SRI shown in Appendix C, a median correlation of 0.5 is "moderate" by definition. A rigorous evaluation of the influence of using SPEI rather than other drought indices (of which there are many—see WMO 2016 reference) would require computing and comparing glacial/nonglacial versions of all those indices as well. For reasons of data availability and scientific suitability to the problem, as already outlined in the manuscript, such a computation is neither scientifically desirable nor practical.*

RC: *Third, a big problem of this manuscript is that it does not take into account the CO2 emissions during the*

*processes of melting glaciers. If just considering melting glaciers as the water source of runoff but ignoring the fact that it is also the emission source of greenhouse gas, the whole conclusion will be embellished. The authors should pay attention to this issue.*

AR: *Please see our previous Author Comment regarding the relative contribution of glacier melt to the global carbon cycle. We have made no changes in response to this comment.*

RC: ***Fourth, the authors are encouraged to further compare the results under the RCP 4.5 scenario against those under the RCP 8.5 scenario, although there are some relevant sentences in Lines 222-232. The potential findings may provide a reference for better policy making of drought mitigation.***

AR: *We have added text to the Abstract, Discussion, and Conclusions to highlight differences between RCP 4.5 and 8.5. We have also added counts of how many basins have positive buffering on each metric for both RCP 4.5 and 8.5, as suggested by another reviewer. Example text additions include:*

> ***Abstract***
> *... Results are similar under RCP 4.5 and 8.5 emissions scenarios, though results for the higher-emissions RCP 8.5 scenario show wider multi-model spread (uncertainty) and greater incidence of decreasing buffering later in the century.*
>
> ***Conclusions***
> *... Drought buffering persists under both RCP 4.5 and RCP 8.5 emissions scenarios, but with greater uncertainty and more decline over time in the RCP 8.5 scenario. Thus, the reliability of glacial drought buffering in the 21st century depends on whether the world acts to mitigate greenhous gas emissions.*

*We are hopeful that these additions will support better policy-making, as the reviewer suggests. Thank you for the encouragement.*

**1.2. Specific comments**

RC: ***1. The authors are encouraged to summarize the innovation/major contributions of this study in the last paragraph of the Introduction part.***

AR: *We have summarized the major contributions of the study in the Abstract, Discussion, and Conclusions. We prefer not to interrupt the reading flow of the manuscript from the Introduction through to the Methods, so we have not added the suggested summary there.*

RC: ***2. Eight GCMs are used in this study. Which eight GCMs? The authors do not mention them in the text. Please clarify this clearly.***

AR: *We have added a list of the eight GCMs considered in Section 2. Thank you for pointing out their inadvertent omission in the previous version.*

RC: ***3. Lines 86-87: "... we use a relatively short 3-month integration timescale, which is typical of that used to assess streamflow drought." Any references for this statement?***

AR: *We have added two example citations: López-Moreno 2013 [1] and Peña-Gallardo 2019 [2].*

RC: ***4. As mentioned in Line 243, "current glacier meltwater production is unsustainably high". I would suggest the authors discuss more about the possible results under such situation.***

AR: *We have modified the final sentence of the paragraph to clarify that our results already incorporate those findings. The phrase "unsustainably high" is taken from the Rowan et al (2018) and Pritchard (2019) studies and refers to the ongoing retreat of glaciers in high-mountain Asia, which is reflected in the glacial runoff modelling that enters our SPEI calculation.*

RC: **5. Appendix A: the authors only show the results for all the 56 basins in Figure A1, but do not provide in-depth discussions/explanations. For example, for some basins, blue shades and orange shades are overlapped; for others, blue shades and orange shades are not overlapped. Is it consistent with the result of K-means cluster analysis? Moreover, it is better to provide sufficient physical interpretations.**

AR: *We have presented representative example basins in the main text and given physical interpretation there. Giving specific interpretation for each of the 56 basins would require adding a great deal of text, distracting from the key findings of the manuscript. We have chosen to keep the manuscript concise—a choice that was praised by Reviewer 3 in their assessment. All data and code for the analysis is available to readers who are interested in specific basins.*

**2. Reviewer 2, Caroline Clason**

**2.1. General comments**

RC: *This study, by Ultee and co-authors, uses the results of existing GCM simulations and global glacier modelling to assess the importance of glacier runoff for drought buffering in the SPEI (Standardized Precipitation-Evapotranspiration Index). The SPEI is modified to quantify drought buffering by glacier runoff for 56 glacier catchments and compared with a baseline that does not include glacier runoff. The rationale for this work is outlined clearly within the introduction, and one of the key findings - that glacial drought buffering might extend beyond the end of the century despite passing glacier basin peak water for many regions - is a fascinating one that warrants further study. Overall, I found this to be a well-written and well-devised study that will be of benefit to researchers working on glacier-fed water security, and I identify no major issues with the manuscript.*

AR: *We thank the reviewer for her attention to the manuscript. We appreciate her positive assessment and helpful comments.*

**2.2. Specific comments**

RC: *Abstract - I think the abstract currently does a good job of explaining the "bare bones" of the study, but would benefit from some additional overview of the methods employed and key outcomes.*

AR: *Added the following sentences to better describe the methods and key outcomes:*

> *We compute one baseline version of SPEI and one version modified to account for glacial runoff changing over time, and we compare the two for each of 56 large-scale glaciated basins worldwide. ... When glacial runoff is included in SPEI, the number of droughts is reduced in 40 of 56 basins and the average drought severity is reduced in 53 of 56 basins, with similar counts in each time period we study. ... Results are similar under RCP 4.5 and 8.5 emissions scenarios, though results for the higher-emissions RCP 8.5 show wider multi-model spread (uncertainty) and greater incidence of decreasing buffering later in the century. A k-means clustering analysis shows that glacial drought buffering is strongest in moderately glaciated, arid basins.*

RC: *Line 110 – This is probably just something that I've seen and thus can never unsee, but you could consider changing $SPEI_W$ to $SPEI_G$ as currently it looks a bit like 'spew' and 'G' seems more representative.*

AR: *Ha! Yes, changed throughout.*

RC: *Line 106 – could the title of section 2.1 be a little more snappy?*

AR: *Shortened to "SPEI modified to account for glacial runoff", to reflect the main focus of the section. The points "basin total" and "variable stomatal conductance", previously included in the section title, are described in the detailed methods in Appendix B. Thank you for this suggestion.*

RC: *Line 144-145 – I'd suggest writing out the four variables here for clarity rather than having to refer to Table 1 which doesn't appear for a few pages.*

AR: *Yes, good point. We have revised to refer back to Section 2.4 rather than ahead to Table 1. The relevant lines now read*

> *2.4. Change in drought buffering over time*
> *... Finally, we compute the Spearman rank-correlation coefficient between the multi-GCM mean of each statistic for each basin and several basin-scale variables that could account for differences in buffering: the percent glacier coverage by area, the total basin area, the historical mean precipitation over the basin, and the aridity index of the basin over the historical period (see Table 1). The aridity index employed here is the ratio of multi-GCM mean precipitation divided by multi-GCM mean PET, for each basin, over the period 1980-2010.*
> *2.5 K-means cluster analysis of basin characteristics that affect buffering*
> *... We then explored the differences between clusters using the four potentially explanatory variables described in Section 2.4 above.*

**RC:**    *Lines 147, 169, and 188 – This could be down to personal preference, but while I appreciate the idea of using headline results as section headers, I wonder whether this would be better changed to a description of the wider content in each section. E.g. "Impact of glacier runoff for basin water supply"; "Glacier runoff influence on drought severity and frequency", and "Influence of glacier cover for drought buffering".*

*AR:*    *We tested out this suggestion, but it was tough to land on an appropriately general headline to capture section 3.3. It is true that we describe the influence of glacier cover (and aridity index) for drought buffering as the key result, but in the k-means clustering we tested several other variables as well. Hmm. We've kept the original headers for now. With luck, they will help time-crunched readers skim through the key results too.*

**RC:**    *Figure 1 – I'd include the region / country name for each panel for ease of comparison. I also wondered whether it might be worth exploring changing one (or both) of the colours, as where they overlap it produces a brown-ish colour that's not so dissimilar to that for the "without glacier runoff" scenario. Maybe changing the colour would still result in a bit of a "muddy" overlap, in which case stick with what you've got.*

*AR:*    *We have added a region abbreviation to each panel in Figure 1 and A1—full region or country names would be poorly legible. We have also brightened the $SPEI_N$ colour so that it is more distinct from the muddy brown in the overlapping areas. Finally, we have increased saturation on both colours for better readability. Thank you for these suggestions.*

**RC:**    *Figure 4 – the blue and green colours here are quite dull on my screen (the median gets particularly lost in the blue box plots), so might warrant brightening up for ease of viewing.*

*AR:*    *We have brightened this figure and reduced the saturation on the box plots to make the median more clear. Thank you for pointing this out.*

**RC:**    *Figure A1 – as for Figure 1, I'd suggest including region names within each panel, and/or consider grouping the panels in this large figure by region, to allow easy intra-region comparison.*

*AR:*    *We have added a region abbreviation to each panel, as in Figure 1. The panels are currently in descending order of total glacier area, which matches their order in the supplementary material of Huss & Hock 2018, so we prefer to maintain their order. It is worth noting that we did preliminary analyses grouping the basins by region, and plotting buffering against latitude, and there were no notable relationships there. This is another reason we have not prioritised intra-region comparison in the current manuscript.*

**RC:**    *Lines 293-312 – because the SPEI is such a central component of the methodology here, I'd consider moving this detail into the main manuscript methods.*

*AR:*     *We did try this, but concluded that the long methods section might dissuade some readers. For now we propose to leave the SPEI methods in Appendix B, with the brief overview of the computation remaining in Section 2, and add a sentence to refer interested readers directly to Appendix B.*

**RC:**     *Lines 330-342 – as comment above.*

*AR:*     *See response above - we propose to keep the detailed methods in the appendix and refer interested readers directly there. We are hoping that this format will encourage a broader audience to read the key points of the methods in the main text overview, rather than skipping a section that looks overly detailed.*

**3. Reviewer 3, Sarah Hanus**

**3.1. Overview**

**RC:** *This paper investigates drought buffering by glacier runoff in the 21st century in 56 major river basins worldwide. Glacier runoff can be an important buffer to interannual variability in runoff. However, dynamic glacier runoffs are not included in GCMs and thus also neglected when calculating drought indices (e.g. SPEI). The authors modify the SPEI to consider glacier runoff as part of water accumulation. Using the SPEI, they calculate number and severity of drought events from 1980-2100 using 8 GCMS and RCP 4.5 and RCP 8.5. They compare differences of the modified SPEI including glacier runoff to normal SPEI. Moreover, they perform a correlation and clustering analysis to identify the explanatory variables for drought buffering by glacier runoff. The results suggest an increased water supply and a buffering of drought frequency and severity when accounting for glacier runoff. A large glaciated fraction and low historical precipitation lead to high drought buffering by glaciers as identified by the clustering analysis.*

**3.2. General comments**

**RC:** *This is a very well-written paper focusing on the relevance of glacier runoff for future hydrological droughts worldwide. I enjoyed reading it because it is well structured, concisely written with self-explanatory headers, and it provides a novel analysis of importance of glaciers. I also appreciate the code and data availability including a Jupyter Notebook to reproduce the results.*

*AR:* *We thank the reviewer for her careful attention to our manuscript. The reviewer raises interesting scientific points and makes several good catches in the minor comments and technical correction. Addressing these points has been very helpful in improving the manuscript. We are also glad that the public repository and Jupyter Notebook guide have been helpful.*

**3.3. Specific comments**

**Methodological Limitations / Uncertainties**

**RC:** *In line 115 you mentioned an important assumption of your approach and claim that you will address this simplification below. However, I did not find where this assumption was further addressed in the discussion or elsewhere. Precipitation in mountain areas (where glaciers are located) tend to be larger than in the rest of the basin due to orographic precipitation, especially in basins, which are considerable arid further downstream (e.g. Amu Darya) (Immerzeel et al., 2020, Extended Data Fig.3a). Assuming an equal distribution of precipitation when calculating the modified moisture source term leads most likely to an overestimation of total precipitation in the basin. Thus, the moisture source term in SPEI will be increased, however this increase is an artefact of the method used and not due to including glacier melt. I miss a discussion about this limitation and how it might have influenced the results and conclusions. I would expect that the influence is larger in basins that are considerable drier in the lowlands than in the mountains.*

*Below is a simplified example to illustrate my point using arbitrary values:*
*Precipitation in Mountains (20% of catchment): 3000mm*
*Precipitation in Lowlands (80% of catchment) 500mm*
*Glacier Area 4%*
*Total precipitation: 0.2\*3000 + 0.8\*500 = 1000 mm*
*In the modification of your precipitation input this would result in a total precipitation of*

*1000\*0.96 + 0.04\*3000= 1080mm.*

*In the model of Huss and Hock (2015) which is used to generate the glacier runoff in Huss and Hock (2018) a precipitation correction factor and a precipitation lapse rate is applied to the input precipitation data under the assumption that precipitation in high elevations is likely underestimated in data sets. The precipitation correction factor is used as calibration parameter for the glacier model. Therefore, precipitation as output from the glacier model is different from the original precipitation data set. So, the runoff from glacier area in form of snowmelt and rain will be higher than the precipitation of the original GCM data on this area. This also leads to an increase in the moisture source term of SPEI that does not directly stem from glacier melt itself. Although glacier area is shrinking, the precipitation input of the glacier model is still used for the initial glacier area. I would appreciate a discussion about the possible limitation: How does the usage of total glacier runoff data fixed to the initial glacier area instead of using only glacier melt data impact your results? As an idea, you could increase the moisture source term of $SPEI_N$ by increasing the precipitation over the glacier area by a default factor (e.g. 1.5 as this is the default factor for precipitation correction in GloGEM (Huss and Hock, 2015)) and look if this impacts your results (Fig. 3)*

$$P_{SPEI_N} = \frac{(A - A_G)}{A}P + \frac{A_G}{A}(1.5)P \tag{1}$$

AR:  *This is an excellent point with a helpful toy example. First, we should mention that the problematic line the reviewer first identified, " Our modified SPEI calculation assumes that both precipitation and glacial runoff are distributed evenly across the drainage basin..." was legacy text from a previous version of the manuscript. That assumption is no longer made in the current version of the manuscript; instead, we perform an area-weighted sum of the variables from each GCM over the grid cells intersecting with the basin outline, as described in Appendix B. This method thus allows variation from grid cell to grid cell, but still misses orographic effects (as most GCMs do). We have revised the relevant line of the methods to state:*

> *We note that Huss & Hock (2018) apply an elevation-dependent correction to the precipitation input data for their model; to avoid introducing any further assumptions, we have not attempted to apply any similar correction to the precipitation term in our modified SPEI calculation.*

*We have not attempted to implement a default precipitation correction in the revised version of the manuscript. To do it correctly, we would prefer to know how this correction factor varies from basin to basin. We therefore avoid introducing a new assumption to the methods, opting instead to add a discussion of this effect to the Discussion section. It is worth noting that the precipitation scaling by Huss & Hock is an attempt to correct a known problem with GCM output. As such, although it was not the main focus of our analysis, we believe that maintaining the precipitation scaling they employ is a co-benefit of our approach. We have added the following paragraph to the Discussion:*

> *Our offline computation method also comes with the caveat that it is likely to capture effects that are not strictly glacial. Because glacial runoff consists of both meltwater from glacier ice as well as precipitation falling on and then running off of glaciers, a consistent comparison of past and future runoff from a currently glacierized basin requires a catchment outline that does not change over time. The runoff output from Huss & Hock (2018) tracks all water discharge from a constant catchment for each glacier simulated, such that snow and rain falling on areas from which glaciers retreat during the 21st century will still be counted as "glacier" runoff (see Appendix B2). A glacier catchment where the initial glacier has vanished would produce no glacial melt, but would still produce runoff comprised of precipitation. In principle, this detail is unlikely to affect our results, since we correct the SPEI moisture source term to avoid double-counting (Equation 1). However, Huss & Hock (2015) also apply a precipitation enhancement factor $c_{prec}$ in their model to account for underestimation of high-elevation precipitation in GCM-derived datasets. The value of $c_{prec}$ is calibrated per glacier starting from a default of 1.5 (see Huss & Hock, 2015, Equation 2 and Section 4). That is, for a catchment where the initial glacier has vanished, GCM-derived precipitation input would still be scaled up by a factor of $c_{prec}$. This scaling may produce $SPEI_G > SPEI_N$ even when the glaciers in a catchment have completely receded. For the most heavily glaciated basin in our analysis, the Copper basin at 20% glaciated, scaling up precipitation by the default $c_{prec}$ value over all glaciated area would result in a 10% increase in total basin precipitation. Thus, the drought buffering effects we present here may be in part attributed to the efforts by Huss & Hock (2015) to capture orographic precipitation enhancement, rather than coming from glacial meltwater alone. We believe that this is in fact an additional benefit of accounting for runoff from glaciated regions in greater detail than the current generation of GCMs permits.*

**RC:** *In general, I missed a bit more in depth discussion about limitations and uncertainties of the study. It was touched upon (l.249- 257) but this is only related to possible feedbacks of glacier retreat that are not captured in the GCMs, but I think there would be more uncertainties to discuss, for example how informative it is to average hydro-climatic variables over the whole basins, and to discuss the possible limitations stated above.*

 *AR:* *We have added to the Discussion as outlined above. Please also see above re: methods - we are no longer averaging hydro-climatic variables over the basins.*

**Basins without drought buffering of glaciers?**

**RC:** *Figure 3: The figure contains a lot of data points. Therefore, it is difficult to grasp, how many basins experience more/less/equal buffering in future and how many basins do not experience a buffering at all in the past and the future (black dots on the zero line). I would suggest to either add these numbers in the figure or in the text describing the figure. In the abstract you claim that accounting for glacier runoff tends to increase drought buffering even in basins with little glacier cover. Looking at Figure 3, I see also a lot of basins without difference especially for small glaciers (black dots close to the zero line). How do you explain this?*

 *AR:* *We have added a count of basins with strictly positive buffering, $N_{>0}$, on each panel of Figure 3 and Figure A2. We have also added a clarification on the x-axis scale (logarithmic) to both Figure 3 and Figure A2.*

*The $N_{>0}$ counts indicate that most basins do have positive buffering, especially of drought severity. Note that the x-axes of Figure 3 and Figure A2 are logarithmic; therefore, even points near the middle of the plot*

*have less than 1% glacier coverage by area. Our claim in the abstract is that accounting for glacier runoff decreases (i.e. buffers) drought frequency and severity, even for basins with relatively little (<2%) glacier cover, and that is supported by the visual and quantitative results presented in Figure 3 and A2. We do not claim a general* increase *in buffering; that finding holds only for the smaller set of basins within Cluster 1, and we do not highlight it in the abstract. Regarding basins with little buffering and little change over time, Figure 3 shows that there is little buffering effect in some basins where glacier coverage is extremely low. This agrees with the cluster analysis. We also note that there is a nonzero tolerance for "no change", such that very small changes in buffering values that were already small would be recorded as "no change".*

**RC:** ***How do you explain that there are quite a lot of basins where you do not see drought buffering in 1980-2010 in Figure 3, although you use the same standardization set for SPEI$_w$ and SPEI$_N$ as explained in l.305-312. I would argue that this should lead to higher SPEI$_w$ values than SPEI$_N$ values in the past and thus maybe to a reduced number and severity of droughts for SPEI$_w$ compared to SPEI$_N$.***

*AR:* *With the counts of basins with positive buffering now added to Figure 3, we see that there are 16 basins without buffering on number of droughts, but only 3 without buffering on drought severity during the historical period. We agree that in general the use of the same standardization set should lead to higher SPEI$_G$ values, but the basins without buffering in the historical period are those with such low glacier coverage that we would expect a minimal glacial contribution to total basin runoff. In those cases, the basin is likely in a precipitation-dominated regime, so we would not expect strong glacial drought buffering.*

**3.4. Minor Comments**

**RC:** ***L.4 I would specify "hydrological drought conditions".***

*AR:* *Added.*

**RC:** ***L.7 can you quantify more what "relatively little" glacier cover means?***

*AR:* *Added general clarification, now reads "relatively little (<2%) glacier cover".*

**RC:** ***Abstract: I would appreciate some more quantification of the results, e.g. for how many basins did the inclusion of glacier runoff buffer droughts?***

*AR:* *Added the following sentence:*

> *When glacial runoff is included in SPEI, the number of droughts is reduced in 40 of 56 basins and the average drought severity is reduced in 53 of 56 basins, with similar counts in each time period we study.*

**RC:** ***L.95: A list of used GCMs in the appendix would be useful for the reader (now you have to search for it until you find it in Figure B3)***

*AR:* *Added a list. Good catch.*

**RC:** ***L 137: A short explanation of the definition of aridity index you use would be helpful. I assume you use ratio of precipitation to evaporation such that high values mean wetter conditions? (compare l.201, l.235)***

*AR:* *The reviewer is correct. We have added this explanation to section 2.4 and added a clarification to the caption of Figure 4.*

**RC:** *Figure 1 and Figure A1: I think adding the percentage of glacier coverage next to the name of each basin would give the reader a better overview of the glaciation of the basin and make it easier to interpret the results of the figure*

*AR:* Added, and we agree this makes interpretation easier. Thank you!

**RC:** *In Figure 3, Table 1 and Figure 4 I miss the description of whether RCP 4.5 or RCP 8.5 was used.*

*AR:* We have added this information to the caption of each. They reflect RCP 4.5. We have also added a line to refer readers to Figure A2 for RCP 8.5 results. Thank you for pointing this out.

**RC:** *Figure 3: no units for the severity are given as axis labels.*

*AR:* Severity is a sum over SPEI, a unitless standardized index. The severity is therefore unitless as well. We have added a sentence to clarify in section 2.3.

**RC:** *Table 1: AI: abbreviation not mentioned before*

*AR:* Corrected caption. Thank you.

**RC:** *L.368: "The only notable difference between the two sample sets is in the magnitude of glacial runoff." This statement is not clear to me, I do not see it in Figure B4. I rather see differences in precipitation between the upper and lower row.*

*AR:* The reviewer is correct. This was a mistake on our part. We have corrected the description and revised surrounding text to explain the difference. The text now reads

> *The only notable difference between the two sample sets is in the magnitude of precipitation. The basin/model combinations with true negative buffering have relatively large precipitation magnitudes and ubiquitously positive P-PET. In heavily glaciated basins ($A_g/A$ large) with periods of low runoff, there may arise an imbalance when $R < \frac{A_g}{A}(P - Pet)$, such that the modified water balance in Equation 1 is lower than the original.*

**RC:** *L. 339 SPEIG should be SPEIW*

*AR:* We've revised to use SPEI$_G$ throughout.

**References**

[1] *J.I. López-Moreno, S.M. Vicente-Serrano, J. Zabalza, S. Beguería, J. Lorenzo-Lacruz, C. Azorin-Molina, E. Morán-Tejeda (2013). Hydrological response to climate variability at different time scales: A study in the Ebro basin. Journal of Hydrology, 477: 175-188. doi: 10.1016/j.jhydrol.2012.11.028.*

[2] *Marina Peña-Gallardo, Sergio M. Vicente-Serrano, Jamie Hannaford, Jorge Lorenzo-Lacruz, Mark Svoboda, Fernando Domínguez-Castro, Marco Maneta, Miquel Tomas-Burguera, Ahmed El Kenawy (2019). Complex influences of meteorological drought time-scales on hydrological droughts in natural basins of the contiguous Unites States. Journal of Hydrology, 568: 611-625. doi:10.1016/j.jhydrol.2018.11.026.*

---

## Author Response (AR2)

**Author Response to Reviews of**

**Glacial runoff buffers droughts through the 21st century**

L. Ultee, S. Coats, J. Mackay
*Earth System Dynamics Discussions*
* * *
**RC:** *Reviewer Comment*,    AR: *Author Response*,    □ Manuscript text

We thank the reviewers for their consideration throughout this process. Only one reviewer requested changes on the most recent version of the manuscript. We respond to each suggestion directly below, where necessary quoting excerpts of revised text. A revised manuscript with changes highlighted is attached

**1.  Anonymous Reviewer 1**

Reviewer 1 did not participate in this round of review.

**2. Reviewer 2, Caroline Clason**

**2.1. General comments**

**RC:** *No suggestions for revision - I would accept as it is.*

AR: *We thank the reviewer for her consideration.*

**3. Reviewer 3, Sarah Hanus**

**3.1. Overview**

**RC:** *Thank you for submitting the revised version of your manuscript. All my minor comments were well addressed. However, some clarifications regarding my comments referring to the methodological limitations / uncertainties are still needed.*

**3.2. Basin-averaged precipitation**

**RC:** *The authors explain that they perform an area-weighted sum of the variables from each GCM over the grid cells in the basin and thus the previous assumption "Our modified SPEI calculation assumes that both precipitation and glacial runoff are distributed evenly across the drainage basin..." does not hold anymore. Can you clarify this? Does an area-weighted sum mean that you weighted the precipitation output of each GCM grid cell in the respective basin by the area of the corresponding grid cell? And the sum gives the mean precipitation of a basin? However, in Equation 1 a fraction of the total precipitation in the basin is replaced by the glacier runoff from Huss and Hock (2018). If I understood your explanation correctly, my first comment remains: There is the possibility of an increased moisture source term in the modified SPEI that does not stem from glacier runoff but from the methodological limitation. This might be the case if glaciers are located in a rather wet part of the basin, e.g. Amu Darya basin (see the toy example in the previous comment). If you would want to circumvent this limitation, you would have to subtract the glacier area fraction from the precipitation amounts of the actual grid cells where glaciers are located and not from the basin-averaged precipitation.*

*I do not expect you to change this methodology, as this might only very slightly impact the results, but I expect some discussion of this potential limitation similar to the discussion of the precipitation correction factor. An example quantification of whether this methodological limitation affects the main results of your study would be helpful to understand it. This quantification could be combined with the following comment as both methodological limitations likely lead to an increased moisture source term in the modified SPEI which does not relate to glacier runoff itself and could potentially affect the results regarding buffering of droughts by glacier runoff.*

**AR:** *Thank you for clarifying this comment further. We agree that our method might overestimate the moisture source when glaciers are concentrated in the wettest parts of the basin, and we have conducted the sensitivity test of moisture overestimation in SPEI as suggested.*

*We should mention that the area-weighted sum of GCM-derived variables gives the **total** (not the mean) of each variable over the basin. The area-weighting multiplies the value at each grid cell by the percent of that grid cell that falls within the basin boundary. For example, consider a basin that covers 5 grid cells: one grid cell in the center is covered entirely, and the basin boundaries cross four adjacent grid cells such that only a portion of each is covered. Say the four adjacent cells are 40%, 30%, 20%, and 10% covered, respectively. Then the area-weighted sum of precipitation would be:*

$$P_{total} = P_1 + 0.4P_2 + 0.3P_3 + 0.2P_4 + 0.1P_5$$

*and the corrected moisture source term would be:*

$$\tilde{P} = \frac{A - A_g}{A} P_{total} + R$$

*Because the area-weighting in the computation of $P_{total}$ is per fraction of grid cell covered, rather than per fraction of basin covered, the arithmetic of assigning glacial runoff to a specific grid cell is less direct than the*

*toy model originally proposed. After puzzling about this for a while, though, we are convinced the reviewer is correct in saying that our method could still overestimate moisture. Please see new section B2.1. We have also added specific mention to the main text Discussion:*

> *Our offline computation method also comes with the caveat that it is likely to capture effects that are not strictly glacial. For example, in Equation 1, we scale the basin-total precipitation by the total non-glaciated area rather than scaling down precipitation from the specific GCM grid cells where glaciers are found. This methodological choice may tend to overestimate the moisture source term when glaciers are found in comparatively wet parts of a basin (see Section B2.1). Further, ...*

**3.3. Precipitation correction factor**

**RC:** ***The added paragraph to the discussion is well written and explains concisely and honestly the potential limitations. Thanks a lot! It also shows for one example basin how much the precipitation would increase assuming a default precipitation factor. What would still be needed to understand the implications of this limitation is the connection between the increase in precipitation and the changes in SPEI. The effect of an increase in precipitation on SPEI is not straightforward. The higher the potential evaporation, the larger the effect of increased precipitation on Di values (Eq. B1). Therefore, the effect will be larger in summer and if the basin is located in lower latitudes/altitudes, I assume. Then, to calculate the SPEI, the values are compared to the standardization set of SPEI. It is not clear to me how much the SPEI and your results would change if there would be a % increase in precipitation in the Cooper basin. I agree that it makes no sense to implement a precipitation correction for all basins in the manuscript. However, it is important to know that the methodological limitations have no significant effect on the results. Therefore, I suggest the authors follow their calculations of the results for two basins, given no glacier melt but an increase in precipitation. One good example would be the basin with the largest glaciation as suggested by the authors already, where the increase in precipitation would be largest. Another example could be a basin with high potential evapotranspiration, e.g. Indus basin (see Fig. 2 in Laghari et al, 2012) where a change in the moisture source term leads to larger Di values. It would be valuable to see if this increase in precipitation changes the SPEI and the amount of droughts / frequency of droughts or whether the results are robust to these methodological limitations.***

**AR:** *Thank you very much for this detailed suggestion. We have added extended discussion of both the precipitation correction and basin-total moisture overestimation in section B2.1 "Non-glacial effects deriving from our processing". We computed a new version of SPEI with a uniform scaling-up of precipitation by a factor of 1.5 in glaciated areas, which we have called "$SPEI_{PS}$". We compare $SPEI_{PS}$ to $SPEI_N$ and $SPEI_G$ in the new Figure B2 (ensemble mean SPEI over time) and Table B1 (buffering statistics).*

*The suggested sensitivity test showed that $SPEI_{PS}$ did overlap with $SPEI_G$ in the Copper basin, and it did have fewer and less severe droughts than $SPEI_N$. However, the effect was readily distinguishable from the glacial drought buffering by difference in magnitude and temporal pattern. In the Rhone, Tarim, and Majes basins, the $SPEI_{PS}$ ensemble means were closer to $SPEI_N$ than to $SPEI_G$. In addition to the description in section B2.1, we have added the following to the main text Discussion:*

> *A sensitivity analysis (Section B2.1; Table B1) indicates that non-glacial enhancement of the moisture source term in SPEI can produce strong drought buffering, but the effect is distinct from the drought buffering calculated from glacial runoff. More detailed analyses to partition the two effects will improve future forecasts of glacial drought buffering.*

---

## Author Response (AR3)

**Author Response to Reviews of**

**Glacial runoff buffers droughts through the 21st century**

L. Ultee, S. Coats, J. Mackay
*Earth System Dynamics Discussions*
* * *
RC: *Reviewer Comment*,    AR: *Author Response*,    ☐ Manuscript text

We thank the reviewers for their consideration throughout this process. Only one reviewer requested changes on the most recent version of the manuscript. We respond to the suggestion directly below. A revised manuscript with changes highlighted is attached.

**1. Reviewer 3**

**1.1. Minor revisions suggested**

RC: *Thanks a lot for answering my comments and for the additional analysis to estimate the potential effects of methodological limitations! In my eyes the analysis has shown that the enhancement of the moisture source term can have an impact on the results. Nevertheless, the impact of glaciers remains. I think the added section in the appendix (B2.1) explains very well your additional analysis and shows honestly the potential limitations.*

*I have only one last suggestion for the manuscript: Can you add one to two sentences in the conclusion which reflect the possible methodological limitations (shown in B2.1 and the sentences added to the Discussion)? I think this would be important, so that also readers who do not dive into the Discussion and Appendix, know that the drought buffering is probably not only coming from the glaciers itself but also due to limitations regarding precipitation inputs etc.*

*Since you spend a lot of time and effort to analyse the effects, adding some conclusive remarks in the Conclusion about this should be rather straightforward.*

AR: *Yes, good point. It was an oversight that we did not edit the conclusions before. I suppose we were optimistic to think that readers will want to read the whole thing!*

*We have added three sentences referring to the limitations indicating areas for future improvement. We have separated the conclusions into two paragraphs to aid readability. Inspired by the reviewer's comment, we have also glossed the "GCM" acronym again for the benefit of readers who haven't thoroughly read earlier sections.*

*New sentences read as follows:*

*Our analysis also indicates work that can be prioritized to improve future projections of glacial drought buffering. First, our method can be applied at smaller scale, analysing runoff per glacier catchment rather than aggregating into large-scale basins, which will correct overestimation of moisture when glaciers are in the wettest part of a large basin. Second, our method can be refined to partition glacial effects on SPEI from non-glacial enhancement of moisture (such as parameterized orographic precipitation) in glacier models. Third, more fundamental work ...*